

# Evaluation of atmospheric sulfur dioxide simulated with the EMAC (version 2.55) Chemistry-Climate Model using satellite and ground-based observations

Ismail Makroum [1], Patrick Jöckel[1], Martin Dameris[1], Nicolas Theys[2], and Johannes De Leeuw[3,4]

[1]Deutsches Zentrum für Luft- und Raumfahrt (DLR), Institut für Physik der Atmosphäre, Oberpfaffenhofen, Germany
[2]The Royal Belgian Institute for Space Aeronomy, Belgium
[3]The National Institute of Oceanography and Applied Geophysics, Italy
[4]The Abdus Salam International Centre for Theoretical Physics, Italy

**Correspondence:** Ismail Makroum (Ismail.makroum@dlr.de)

## 1 Abstract

Sulfur dioxide ($SO_2$) is a key atmospheric pollutant, primarily emitted through human activities such as fossil fuel combustion. In atmospheric models, accurate representation of $SO_2$ emission sources, transport, and removal processes are essential for evaluating air quality and radiative forcing.

In this study, we present, for the first time, a comprehensive examination of atmospheric $SO_2$ simulated by the ECHAM/MESSy Atmospheric Chemistry (EMAC) model. First, the tropospheric sulfur budget simulated by EMAC is verified to be close, that is, all sulfur sources and sinks are balanced, ensuring no artificial gain or loss occurs over time due to numerical or conceptual errors. This budget closure is a prerequisite for any further analysis. Second, the results of EMAC simulations are compared with observations from three ground-based networks (the Clean Air Status and Trends Network (CASTnet), the European

Monitoring and Evaluation Program (EMEP), and the Acid Deposition Monitoring Network in East Asia (EANET)), mainly over polluted regions, and with vertical column densities retrieved from a TROPOspheric Monitoring Instrument (TROPOMI) on board the Copernicus Sentinel-5 Precursor mission (Sentinel-5P) satellite. The EMAC simulated $SO_2$ concentrations near the Earth's surface for the year 2019 are, depending on the region, between $1.4$ and $1.8$ times larger than observed. This discrepancy aligns well with the differences between simulated and retrieved satellite-based measurements of $SO_2$ vertical

column densities over the same regions. It indicates that the prescribed $SO_2$ emissions used for the EMAC simulations might be overestimated. Over a longer time period (2000-2019), the EMAC simulation reproduces the measured declining trends of $SO_2$ concentrations and deposited sulfur fluxes in the USA and Europe, but fails to simulate the observed trends in East Asia. This is most likely attributable to the prescribed $SO_2$ emission inventories. Furthermore, sensitivity simulations are performed to assess the emitted amount of $SO_2$ following the Raikoke and Ulawun volcanic eruptions in 2019. The results show a very

good agreement of the simulated temporal evolution of the amount of atmospheric $SO_2$ after the eruptions with that retrieved from satellite-based observations.





## 2 Introduction

Air pollution remains a significant global challenge, affecting both, human health and the Earth's climate (Wood et al., 2024; Arias et al., 2021). Among various pollutants, $SO_2$ plays a key role due to its strong influence on atmospheric chemistry, air quality, and climate processes (Seinfeld and Pandis, 2016; Myhre et al., 2013). Anthropogenic $SO_2$ emissions primarily originate from the combustion of sulfur-containing fossil fuels, oil refining, and metal smelting (Smith et al., 2011; Klimont et al., 2013), while natural sources include volcanic eruptions, the oxidation of dimethyl sulfide (DMS) emitted from the ocean, and minor biogenic contributions from land ecosystems (Lana et al., 2011; Fioletov et al., 2016; Quinn et al., 2011).

$SO_2$ is the dominant precursor of sulfate aerosols, which influence the Earth's radiation balance by scattering incoming solar radiation and acting as cloud condensation nuclei (CCN) (Charlson et al., 1992; Lohmann and Feichter, 2005). These processes contribute to short-term climate cooling, partially offsetting warming caused by greenhouse gases (Arias et al., 2021; Albrecht, 1989). At the same time, $SO_2$ contributes to adverse environmental effects such as acid deposition and impacts on the stratospheric ozone layer (Seinfeld and Pandis, 2016; Solomon, 1999).

While global $SO_2$ emissions have declined in many industrialized regions due to regulatory efforts (Klimont et al., 2013; Liu et al., 2018), emissions remain high in rapidly developing countries like India and China (Dahiya et al., 2020). Furthermore, episodic volcanic eruptions introduce large amounts of $SO_2$ into the atmosphere, affecting its distribution on regional and global scales (Carn et al., 2017). Despite improvements in satellite monitoring and emission inventories, uncertainties remain regarding the atmospheric lifetime, transport, and transformation of $SO_2$ (Wang et al., 2014).

Accurately simulating the complex processes governing $SO_2$ behavior in the atmosphere is essential for understanding its role for air quality, climate forcing, and environmental impacts such as acid rain. Chemistry-climate models (CCMs) like EMAC provide a comprehensive framework to represent emissions, chemical transformations, transport, and deposition of sulfur compounds within the coupled atmosphere system. EMAC, in particular, incorporates detailed tropospheric and stratospheric chemistry schemes, making it well suited to investigate the sulfur cycle from emissions to atmospheric sinks (Jöckel et al., 2010).

The current study investigates the distribution and budget of tropospheric $SO_2$ using the EMAC model and observational datasets. The study evaluates the model's ability to reproduce $SO_2$ spatial and temporal distributions by comparing model simulation results with observations retrieved from a satellite instrument and with ground-based measurements. Furthermore, this paper investigates the tropospheric sulfur chemistry within the employed EMAC model. This is done by examining the $SO_2$ emissions, the sulfur related chemical processes, and the sink processes (including wet and dry deposition, and sedimentation) to verify the model's ability to conserve sulfur mass. This conservation is a prerequisite for the comparative analysis and inter-comparison with results from other models and with observational data, thereby showing the numerically correct representation of chemical processes simulated within the model.

This paper includes the following sections: The used EMAC model setup is illustrated in Sect. 3. A detailed study of the tropospheric sulfur budget in the EMAC model is presented in Sect. 4. Sect. 5 shows the evaluation of the simulated global distribution of $SO_2$, as well as the variations of $SO_2$ following eruptive volcanic events, using data retrieved from the TROPOMI



instrument. Sect. 6 assesses the comparison of simulated $SO_2$ concentrations and sulfur deposition fluxes at the Earth's surface with ground-based measurements. Last but not least, the Conclusions and an Outlook of this study are presented in Sect. 7.

## 3 Model description

### 3.1 The EMAC model

In the present study, a detailed investigation and evaluation of $SO_2$ simulated by the global EMAC model Jöckel et al. (2016) integrated within the Modular Earth Submodel System (MESSy) framework (Jöckel et al., 2010), is undertaken. The EMAC model is a comprehensive global CCM that represents physical and chemical processes in the troposphere and middle atmosphere, along with their interactions with the land surface, ocean systems, and human-induced changes such as emissions and land-use (Jöckel et al., 2010, 2016). The second version of the Modular Earth Submodel System (MESSy2) and the 5th gener-

ation European Center Hamburg general circulation model (ECHAM5) (Röckner et al., 2006) make up the EMAC model used in this study. The physics-related submodels within the MESSy framework have been adapted from the physics routines of ECHAM5 (Jöckel et al., 2016). Only the spectral dynamical core, the flux form semi-Lagrangian (FFSL) large scale advection scheme (Lin and Rood, 1996), the time integration loop, and the Newtonian relaxation methods retain their original structure from the ECHAM5 base model.

The results analysed here stem from the RD1SD-base-01 EMAC simulation (Jöckel et al., 2024) that has been performed under the CCMI-2022 protocol (CCMI, 2023). Here, the emissions of $SO_2$ and other sulfur species are caclulated by the submodels OFFline EMISsions submodel (OFFEMIS) (formerly called OFFLEM) for prescribed emission fluxes (Kerkweg et al., 2006b), and AIRSEA (calculating the air-sea exchange of chemical species Pozzer et al. (2006)). Details about the emission setup used here are described in a separate subsection (Sect. 3.2).

Chemical reactions in the gas phase are computed by the submodel Module Efficiently Calculating the Chemistry of the Atmosphere (MECCA) (Sander et al., 2019), while the Scavenging Submodel for Regional and Global Atmospheric Chemistry Modeling (SCAV) simulates the aqueous phase kinetics and scavenging processes in the atmosphere (Tost et al., 2006). The dry deposition of gases and aerosols is calculated by the dry deposition submodel (DDEP) (formerly called DRYDEP) (Kerkweg et al., 2006a), and aerosol sedimentation is calculated by the aerosol sedimentation submodel (SEDI) (Kerkweg et al., 2006a).

The sampling along sun-synchronous satellite orbits submodel (SORBIT) to sample model results on-line along orbits of sun-synchrinously orbiting satellites, as described by Jöckel et al. (2010), has been applied to facilitate a direct comparison between simulated Vertical Column Density (VCD) of trace gases such as $SO_2$, with observations from satellite instruments.

The RD1SD-base-01 EMAC simulation results analysed in this study cover the years 1970 to 2019. The simulation was performed at a resolution of T42L90MA with output of results every 5 hours of the simulated period. The spectral resolution

(triangular truncation) T42 corresponds to a quadratic Gaussian horizontal grid of roughly $2.8° \times 2.8°$ in both, longitude and latitude coordinates, and L90 denotes 90 vertical layers (with a median lowest level height of 60 m) between the surface and the uppermost model layer centered around 0.01 hPa (Jöckel et al., 2010). For the RD1SD-base-01 simulation, the gas phase chemistry is calculated throughout the entire atmosphere using the Mainz Isoprene Mechanism (MIM1) based on Pöschl et al.





(2000). This mechanism accounts for hydrocarbons up to 4 carbon atoms, along with isopren (5 carbon atoms). However,
the simulation did not involve an interactive aerosol submodel. Therefore, aerosol effects were just prescribed in both, the
troposphere and the stratosphere, to consider their impact through heterogeneous chemistry and radiative forcing (Jöckel et al.,
2016). To allow for a direct comparison of the simulation results, in particular chemical tracers, between the simulated and
observational data, the RD1SD-base-01 simulation was operated in "specified dynamics" (SD) mode, for which the prognostic
variables temperature, divergence, vorticity and the logarithm of surface pressure were "nudged" by Newtonian relaxation
towards the fifth generation of European Centre for Medium-Range Weather Forecasts (ECMWF) reanalysis data (ERA5
(Hersbach et al., 2020)). The model dynamics of the SD simulations are then aligned with the observed dynamics, aiming a
good reproduction of real meteorological situations.

## 3.2  Description of the used sulfur emissions

Sulfur emissions of both, anthropogenic and natural sources, need to be taken into account. The following prescribed emission
inventories were used:

–  Throughout this study the Coupled Model Intercomparison Project Phase 6 (CMIP6) inventory is selected as the standard
inventory for global EMAC simulations, since it was recommended by the experimental protocol for participation in the
CCMI-2022 model intercomparison initiative (Eyring et al., 2016). The CMIP6 inventory has a horizontal resolution of
$0.5° \times 0.5°$ and it primarily combines bottom-up inventories to provide emission data for climate models. Bottom-up
inventories involve estimating emissions based on detailed data about specific sources and activities, such as energy
consumption and industrial processes. The CMIP6 inventory contains historical emissions from 1850 to 2014, provided
by the Atmospheric Chemistry and Climate-Model Intercomparison Project (ACCMIP) developed by Lamarque et al.
(2010). The historical data are then combined with the shared socioeconomic pathways (SSPs) for projected future
emissions from the IPCC Sixth Assessment Report (AR6) (Calvin et al., 2023). The SSPs used within the CMIP6
inventory provide a range of future scenarios based on varying levels of greenhouse gas emissions and societal changes,
such as SSP1-1.9, SSP1-2.6, SSP2-4.5, SSP3-7, SSP4-6, and SSP5-8.5 (Riahi et al., 2017). These SSPs present different
emission scenarios, in order to explore different future climate outcomes based on varying levels of greenhouse gas
emissions and societal changes. For the RD1SD-base-01 simulation the SSP2-4.5 scenario has been used to prescribe
trace gases emissions, including $SO_2$ emissions after 2014. The SSP2-4.5 is a middle-of-the-road scenario with moderate
emissions, leading to a radiative forcing of $4.5\,\mathrm{W\,m^{-2}}$ by 2100 (Riahi et al., 2017).

–  The terrestrial Dimethyl Sulfide (DMS_terrestrial) emissions are based on the global inventory developed by Spiro
et al. (1992). This inventory was mainly developed to examine gaseous sulfur emissions. Over the years, this inven-
tory has been evaluated by other studies, such as Chin et al. (2000); Vallina and Simó (2007), and Lana et al. (2011).
DMS_terrestrial emissions originate from both, vegetation and soils, and are available as a monthly resolved annual
climatology at a resolution of $1° \times 1°$ (Bates et al., 1987).



**Table 1.** Parameters of the AeroCom explosive and continuous volcanic emissions. $V_T$ (volcano top) corresponds to the altitude of the top of the volcano.

|  | Time resolution | Injection altitude | AeroCom Flux [Tg(S)/a] |
|---|---|---|---|
| Explosive volcanoes | yearly | From $(V_T + 500 \text{ m})$ until $(V_T + 1500 \text{ m})$ | 2.0 |
| Continuous volcanoes | yearly | From $(0.67 \cdot V_T)$ until $(1.0 \cdot V_T)$ | 12.6 |

- Volcanic sulfur emissions from both, continuously degassing and explosive volcanoes are represented by an inventory of the Aerosol Inter Comparison (AeroCom) project as a zonal mean climatology (Dentener et al., 2006). Volcanic sulfur is emitted as 97.5% $SO_2$ and 2.5% $SO_4$. The data are based on the bottom-up Global Emissions Inventory Activity (GEIA) for the years 1750 and 2000 (Andres and Kasgnoc, 1998). Continuously degassing sulfur in the AeroCom inventory is
equally distributed over the grid points with GEIA volcano locations and amounts to a multi-annual total emission of 12.6 Teragrams of Sulfur per year (Tg(S)/a) over all the years (Dentener et al., 2006). The height of these emissions is defined in the upper third of the volcano altitudes, simulating the degassing processes that occur predominantly at the volcano flanks. Explosive volcanic emissions are quantified at approximately $2\ Tg(\text{S})/\text{a}$ over all the years. This estimation is based on the Aerosol Index (AI) provided by the Total Ozone Mapping Spectrometer (TOMS) satellite
sensors (Dentener et al., 2006). The emissions data are distributed evenly across grid boxes that include volcanoes, which were active in the last century (Halmer et al., 2002). It is important to note that these emissions are treated as being continuously released rather than episodic, due the fact that only about one-third of such emissions occur during violent explosive events (Dentener et al., 2006). Furthermore, these emissions are typically defined to occur between 500 and 1500 meters above the peaks of the volcanoes, to accurately represent their dispersal in the atmosphere. The
injection height, time resolution and the sulfur flux of the different volcano types are listed in Table 1.

In addition to the prescribed emissions, sulfur from oceanic Dimethyl Sulfide (DMS_airsea) and from Carbonyl Sulfide OCS are calculated using the submodels AIRSEA (for gas exchange between air and sea) (Pozzer et al., 2006) and TNUDGE (Kerkweg et al., 2006b) for Newtonian relaxation towards prescribed mixing ratios, respectively.

Since the concentration simulated by the model is affected by the prescribed emissions, it is important to understand the
differences between the used CMIP6 emission inventory and other emission inventories. In this study, the Emissions Database For Global Atmospheric Research (EDGAR) emission inventory (Solazzo et al., 2021) is used for this comparison (See Sect. 6). Same as for the CMIP6 inventory, EDGAR is also considered a bottom-up inventory, but with a finer horizontal resolution on grid-maps at $0.1°$ x $0.1°$. EDGAR is developed using a bottom-up approach combining internationally available statistics on activity data with emission factors derived from scientific literature and guidelines (e.g., IPCC) (Crippa et al., 2019). The data is
available as yearly and monthly mean and is emitted into 7 vertical tropospheric levels (0, 20, 92, 184, 324, 522 and 781 meter), as described by Bieser et al. (2011). Version 5.0 of EDGAR (EDGAR5) contains solely histrotical data about anthropogenic





emissions from different sectors such as fossil fuels, agricultural waste burning, ships and roads, starting from 1970 till present. Other emissions from large scale biomass burning, forest fires and sources from land-use forestry are excluded (Crippa et al., 2019). Here, the EDGAR 5.0 inventory is soley used for inter-comparison to provide an estimate of the uncertainty of the magnitude and variability of $SO_2$ emissions over time from the CMIP6 emission inventory.

## 4  Tropospheric sulfur budget simulated with EMAC

This section provides a comprehensive evaluation of the global tropospheric sulfur budget in the RD1SD-base-01 EMAC simulation by examining prescribed sulfur emissions and the removal of sulfur-containing species via deposition over the years 2010 to 2019. These years were selected based on the availability of corresponding observational datasets used for later evaluation. The goal is to verify the internal consistency of the model's sulfur budget: the sulfur emitted into the atmosphere must either remain in the atmosphere (as part of the sulfur burden) or be removed through deposition processes. This ensures that the model conserves mass and accurately represents the sulfur cycle. The principle can be formulated as follows for each year:

$$\Delta B(t) = E(t) - D(t) \tag{1}$$

Here, $\Delta B(t) = B(t_{\text{end-of-the-year}}) - B(t_{\text{start-of-the-year}})$ is the annual change in the atmospheric sulfur burden (in units of mass), $E(t)$ is the total sulfur emission over the year, and $D(t)$ is the total sulfur deposition over the same year. All quantities are integrated over the year. In this context, the burden $B(t)$ represents the total mass of sulfur in the atmosphere (summed over all sulfur-containing species in the model domain) at a given time $t$. The difference $\Delta B(t)$ reflects the net accumulation (or loss) of sulfur in the atmosphere over the year.

Prescribed sulfur emissions, as applied in EMAC, arise from both, anthropogenic and natural sources. In the present study and for the year 2010, fossil fuel consumption, DMS from the ocean (denoted as DMS_airsea), DMS from terrestrial sources (denoted as DMS_Terrestrial), volcanic activity, and maritime shipping collectively contribute to nearly 95% of the sulfur emissions released into the EMAC model atmosphere. Other sources, such as OCS, agricultural waste burning, and road emissions, constitute the remaining 5% of the emitted sulfur.

The released sulfur from these sectors becomes oxidized and is removed from the atmosphere through dry deposition, sedimentation (of sulfuric particles), and wet deposition/scavenging, which rinses sulfur through convective and large scale precipitation (cv+ls). The emitted, deposited and remaining sulfur species are exemplarily examined for the year 2010 in Table 2.





**Table 2.** Detailed list of the emitted and deposited sulfur species for the year 2010 in the EMAC model. The first column represents the sulfur emission sectors and the third column (Tracers) shows the sulfur species deposited within the EMAC model. Suffixes _cs and _l denote species in coarse mode aersol and liquid phase, respectively.

| Type of emissions | | Tracers | Dry deposition | Scavenging (cv+ls) | Sedimentation | Change of burden |
|---|---|---|---|---|---|---|
| Emissions in $Tg(\mathrm{S})$/year | | Depositions in $Tg(\mathrm{S})$/year | | | | |
| Fossil fuels | 50.46 | $SO_3^-$_cs | $7.23E-13$ | | $2.88E-12$ | $1.63E-13$ |
| Awb | 0.05 | $HSO_4^-$_cs | 0.85 | | 4.031 | 0.1319 |
| Aircraft | 0.13 | $CH_2OHSO_3^-$_cs | 0.08 | | 0.3 | 0.013 |
| Ships | 4.92 | $SO_5^-$_cs | 0.001 | | 0.003 | $2.57E-04$ |
| Road | 1.78 | $HSO_5^-$_cs | 0.03 | | 0.13 | 0.004 |
| Biomass burning | 1.06 | $SO_4^{2-}$_cs | 1.07 | | 4.65 | 0.17 |
| Volcanoes | 14.88 | $SO_3^{2-}$_cs | $3.07E-05$ | | $1.44E-04$ | 1.91E-06 |
| OCS | 0.22 | $SO_4^-$_cs | 2.61E$-12$ | | 1.19E-11 | $6.67E-13$ |
| DMS terrestrial | 0.91 | $HSO_3^-$_cs | 0.002 | | 0.009 | $3.90E-04$ |
| DMS airsea | 28.95 | $SO_4$_res_cs | | | | |
| | | $SO_2$_l | | $3.19E-05$ | | $4.39E-07$ |
| | | $H_2SO_4$_l | | $2.77E-05$ | | $7.83E-05$ |
| | | $SO_3^-$_l | | | | $4.92E-14$ |
| | | $HSO_4^-$_l | | 14.15 | | 0.06 |
| | | $CH_2OHSO_3^-$_l | | 2.19 | | 0.01 |
| | | $SO_5^-$_l | | 0.03 | | $1.66E-04$ |
| | | $HSO_5^-$_l | | 2.09 | | 0.002 |
| | | $SO_4^{2-}$_l | | 39.42 | | 0.08 |
| | | $SO_3^{2-}$_l | | 0.002 | | $1.22E-06$ |
| | | $SO_4^-$_l | | $6.38E-12$ | | $3.72E-13$ |
| | | $HSO_3^-$_l | | 0.08 | | $2.20E-04$ |
| | | $OCS$ | | | | 0.06 |
| | | $SO_3$ | | | | $-6.86E-07$ |
| | | $SO_2$ | 21.93 | | | $-0.08$ |
| | | $H_2SO_4$ | 3.56 | | | 0.004 |
| | | $CH_3SO_3H$ | 7.86 | | | 0.01 |
| | | $DMS$ | | | | $-0.007$ |
| | | $DMSO$ | 0.15 | | | $-4.47E-05$ |
| | | $CH_3SO_2$ | | | | $-1.68E-07$ |
| | | $CH_3SO_3$ | | | | $3.09E-04$ |
| | | $S$ | | | | $-4.11E-16$ |
| | | $SH$ | | | | $3.63E-10$ |
| | | $SO$ | | | | $-1.37E-05$ |
| | | **Sum** | **35.54** | **57.98** | **9.14** | **0.48** |
| Total emissions | 103.39 | Total deposition and burden | 103.15 | | | |
| Total emissions - (Total depositions + change of burden) = 0.24 | | | | | | |





Following the same analysis done for the year 2010, an evaluation of the sulfur budget for the years between 2010 and 2019
is shown in Table 3. Here, the tropospheric sulfur budget is nearly perfectly closed with a value near 0 (sulfur deficit) for the
other years as well. In other words, this shows that the tropospheric sulfur budget is effectively balanced, accounting for the
contributions of various sulfur species and their interactions over the specified time period.

**Table 3.** Sulfur budget closure in $Tg(\mathrm{S})/\mathrm{year}$ between 2010 and 2019 in the EMAC model.

| Years | Sulfur emissions in $Tg(\mathrm{S})/\mathrm{year}$ | Total sulfur deposition in $Tg(\mathrm{S})/\mathrm{year}$ | Change of burden in $Tg(\mathrm{S})/\mathrm{year}$ | Sulfur deficit in $Tg(\mathrm{S})/\mathrm{year}$ |
|---|---|---|---|---|
| 2010 | 103.39 | 102.67 | 0.48 | 0.24 |
| 2011 | 102.87 | 102 | 0.50 | 0.37 |
| 2012 | 102.66 | 101.5 | 0.60 | 0.56 |
| 2013 | 102.25 | 101.03 | 0.62 | 0.60 |
| 2014 | 101.41 | 100.22 | 0.61 | 0.58 |
| 2015 | 95.22 | 94.33 | 0.54 | 0.35 |
| 2016 | 92.96 | 92.11 | 0.53 | 0.32 |
| 2017 | 91.03 | 90.15 | 0.52 | 0.36 |
| 2018 | 88.95 | 88.13 | 0.51 | 0.31 |
| 2019 | 86.99 | 86.13 | 0.52 | 0.34 |

Following the examination of the sulfur budget closure in the EMAC model, a comparison with other atmospheric chemistry
models and relevant literature data helps assessing the model performance and identification of potential areas for improvement
(see next Sect. 4.1).

## 4.1 How does the tropospheric sulfur budget in the EMAC model compare to that of other atmospheric chemistry models?

In their studies, Stevenson et al. (2003) and Penner et al. (2001) address the tropospheric sulfur budget within their atmospheric
chemistry models for the year 1990. Stevenson et al. (2003) used the STOCHEM-Ed model, which is a global three-dimensional
Lagrangian Chemistry-Transport Model (CTM), while the IPCC Third Assessment Report (IPCC AR3, Penner et al. (2001))
incorporates the average results from 11 distinct models providing a comprehensive overview of the global tropospheric sulfur
budget. Since the RD1SD-base-01 EMAC simulation covers the years 1970 to 2019, we directly compare the year 1990, which
is used in both references. Note that in Stevenson et al. (2003), sulfur anthropogenic emissions were presented as a single sector,
unlike the detailed breakdown presented in our study, which categorizes the prescribed emissions into sectors such as fossil
fuels, aircraft and shipping emissions, among others. Additionally, the models in both studies applied an interactive aerosol
model, unlike the EMAC model for the RD1SD-base-01 simulation. Therefore, some sulfur species present in Stevenson et al.
(2003) have not been considered in the EMAC model, such as the Methane Sulphonic Acid (MSA) and vice versa (such as
OCS).

For a clear comparison of the processes used by Stevenson et al. (2003) with those in EMAC, Table 4 provides the values for
both, the prescribed emission and the deposition rates, for the year 1990. Despite discrepancies arising from the applied emis-





sion inventories between EMAC and the literature, as well as differences in used chemical reactions and physical processes, this comparative analysis reveals a good agreement between our model results and existing literature.

**Table 4.** A comparison of emission/deposition fluxes in Tg(S)/year between EMAC, STOCHEM-Ed model and IPCC AR3 for the year 1990.

| Sulfur emission sectors / deposition processes | EMAC model | STOCHEM-Ed model | IPCC AR3 |
|---|---|---|---|
| Fossil fuels | 60.29 | 71 | 76 |
| Awb | 0.063 | | |
| Aircraft | 0.00015 | | |
| Ships | 3.075 | | |
| Biomasse burning | 0.969 | 1.4 | 2.2 |
| Volcanoes | 14.88 | 9 | 9.3 |
| OCS | 0.15 | | |
| DMS_terrestrial | 0.901 | 1 | 1 |
| DMS_airsea | 28.1 | 15 | 24 |
| **Total emissions** | **111.13** | **97.40** | **112.50** |
| Wet deposition | 51.03 | 58.2 | 57 |
| Dry deposition | 42.76 | 37.1 | 39.5 |
| Sedimentation | 8.02 | | |
| **Total depositions** | **101.81** | **95.3** | **96.5** |

Consequently, the magnitudes of sulfur emissions across different sectors are consistent between EMAC and the compared models. The total emissions from EMAC align perfectly with those reported in the IPCC AR3, with a slight difference of about

1 Tg of sulfur per year in EMAC. Regarding the STOCHEM-Ed model, the EMAC simulation utilises a larger sulfur emissions by approximately 12%, related to the applied emission inventories. This overall good agreement in magnitude is also apparent in the deposition rates. In EMAC, the deposited sulfur through wet and dry processes is 93.79 Tg(S)/year compared to 95.3 Tg(S)/year and 96.5 Tg(S)/year for the STOCHEM-Ed model and the IPCC AR3, respectively. The remaining deposited sulfur from sedimentation is present only in EMAC and not in the citet literature.

This evaluation highlights the validity of the EMAC model in comparably capturing sulfur emission and deposition rates, despite minor differences in the chemical species and mechanisms used by the compared models. This discrepancy is attributed to the sulfur emissions inventories utilized as input for the models, which play an important role in shaping both, sulfur mixing ratios and deposition processes. Further refinement and validation of these emissions inventories may help to improve the models performance in representing the tropospheric sulfur cycle.

**5  Comparison of SO$_2$ simulated by the EMAC Model with TROPOMI data**

The primary objective of this section is to conduct a comprehensive global-scale comparison between the EMAC simulations and retrievals based on data from the TROPOMI instrument on board the Sentinel-5P satellite, which is the first Copernicus mission specifically designed for atmospheric monitoring, as mentioned by ESA (2017).





TROPOMI is performing atmospheric measurements, particularly for the quantification of VCDs of various gases and
aerosols (Veefkind et al., 2012). These include ozone, formaldehyde, nitrogen dioxide, carbon monoxide, methane, aerosols,
and sulfur dioxide, which is of specific importance for this study. Additionally, TROPOMI, notable for being the first satellite
instrument to measure $SO_2$ columns with the highest spatial resolution (3.5 km by 7 km) among other satellites, serves as a
pivotal dataset for this study. The analysis is focused on the year 2019, chosen for its status as, at the time of our study, being
the first complete year of data available in both datasets (model and observations).

Performing a direct comparison between the EMAC model results and the TROPOMI/Sentinel-5P datasets presents inherent
challenges due to their different methodologies and resolutions. Therefore, Appendix A2 describes the methodologies adopted
in this study to enable a fair and detailed comparison of $SO_2$ between both datasets. To compare with the TROPOMI-retrieved
VCDs, EMAC's VCDs are calculated by applying the standard/polluted averaging kernel (for the so called "standard case")
labeled "AK_polluted" from the Covariance-Based Retrieval Algorithm (COBRA; Theys et al. (2021)). The COBRA prod-
uct represents an advanced $SO_2$ retrieval technique from TROPOMI. Its improved sensitivity to low $SO_2$ levels enables the
detection of previously undetected emission sources, including weakly emitting volcanoes and power plants (Theys et al.,
2021).

This section is divided into three subsections. A comparison of the EMAC results on the global scale, using the standard
("polluted") AK, against TROPOMI retrievals for the standard case is described in Sect. 5.1. The effects of eruptive volcanoes
on the simulated $SO_2$ total column are discussed in Sect. 5.2. Last but not least, a comparatative evaluation of $SO_2$ from both,
anthropogenic and outgassing volcanic emissions, is detailed in Sect. 5.3.

## 5.1 $SO_2$ Vertical Column Densities in 2019: EMAC vs. TROPOMI

A comprehensive analysis of the global distribution and discrepancies of $SO_2$ VCDs as derived from the RD1SD-base-01
EMAC simulation and from TROPOMI/Sentinel-5P satellite observations for the year 2019 is performed. Figures 1 and 2
show the monthly spatial distribution of $SO_2$ VCDs, with EMAC results on the left and TROPOMI data on the right.

Overall, both datasets display low $SO_2$ VCDs (<0.5 DU) over regions like Western Europe, Africa, and Australia. However,
notable discrepancies appear in areas such as Southern Italy, Northeast China, India, Central America, and Papua New Guinea,
where EMAC simulates persistently higher values (>1 DU) compared to TROPOMI (<0.5 DU) throughout most of the year,
except in June and July. In August, elevated TROPOMI signals over Papua New Guinea are caused by the Ulawun eruption
(Kloss et al., 2021).

In June and July, TROPOMI detects higher $SO_2$ over northern latitudes, particularly following the June 2019 Raikoke
volcanic eruption in Russia (De Leeuw et al., 2021). These enhanced signals are absent in the EMAC simulation due to the
omission of such episodic volcanic emissions in the CMIP6 inventory.





**Figure 1.** Geographical representation of SO₂ VCDs from EMAC, calculated with the standard AK (left panels), against TROPOMI retrievals (right panels) in DU for the first 6 months of 2019. The grey zones represent areas with no valid measurements.





**Figure 2.** Continuation of Figure 1 for the last 6 months of 2019.





To disregard the missing volcanic SO$_2$ emissions from Raikoke and Ulawun in the EMAC simulation, a 10-months average

excluding June and July 2019 is used for the comparison with TROPOMI observations. The resulting differences in SO$_2$ VCDs are shown in Figure 3.

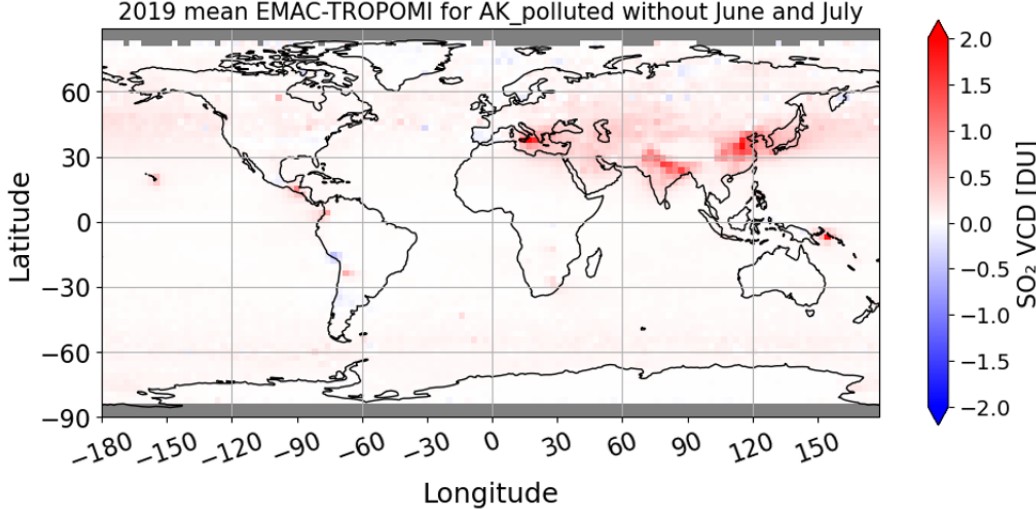

**Figure 3.** SO$_2$ VCD differences between the EMAC model results and TROPOMI retrievals in DU as a 10 months (without June and July) mean.

The comparison between EMAC simulations and TROPOMI observations reveals significant discrepancies in Northeastern China, India, Southern Italy, and at localized sources in the Southern Hemisphere, where EMAC consistently overestimates SO$_2$ VCDs by 1.5 to 2 DU.

Conversely, a better agreement (0.1–0.5 DU differences) is found in Figure 3 over Western Europe, the USA, Africa, Australia, and most of South America, where SO$_2$ emissions are relatively low. The most significant differences arise in regions with large anthropogenic emissions or active volcanoes, such as Etna (Italy), Mt. Fuji (Japan), Ulawun (Papua New Guinea), Nevado Ojos del Salado (Chile), and Kunlun (Tibetan Plateau). Elevated SO$_2$ VCDs in Beijing and India highlight the impact of industrial emissions.

To ensure accurate comparisons, different SO$_2$ sources require appropriate TROPOMI product types and Averaging Kernels (AKs). For instance, AK_15km is used to assess the impact of explosive volcanic eruptions (e.g., Raikoke, Ulawun) on SO$_2$ emissions and deposition, as detailed in Sect. 5.2. For anthropogenic and outgassing volcanic emissions, a relative comparison of SO$_2$ VCD magnitudes is performed, without directly assessing the absolute emitted and deposited SO$_2$ mass from individual point sources.

The differences between the AKs and the correpsonding VCDs are explained in detail in Appendix A2.





## 5.2 Effects of volcanic eruptions on the simulated atmospheric SO$_2$ in the EMAC Model

TROPOMI detected significant SO$_2$ signals in June and July 2019, which were not visible in the EMAC RD1SD-base-01 simulation due to the lack of eruptive events in the applied emission inventories (Figure 3). The Raikoke (48.29°N, 153.25°E) and Ulawun (5°S and 151°E) eruptions injected volcanic SO$_2$ into the stratosphere, increasing stratospheric Aerosol Optical

Depth (sAOD) across both hemispheres (Kloss et al., 2021). Raikoke, the largest SO$_2$ injection into the Upper Troposphere and Lower Stratosphere (UTLS) since Nabro (2011), released about $1.5 \pm 0.2$ Tg (SO$_2$) (Muser et al., 2020; De Leeuw et al., 2021), while TROPOMI estimated 0.14 Tg (SO$_2$) from Ulawun in June and 0.2 Tg (SO$_2$) in early August (Kloss et al., 2021).

To investigate the effects of these two eruptions, we performed additional sensitivity simulations (June - December 2019, initialized with results end of May 2019 of the RD1SD-base-01 simulation), in which the volcanic SO$_2$ emissions were taken

into account with the submodel TREXP (Jöckel et al., 2010). Two different SO$_2$ vertical emission profiles (named StratProfile and VolRes1.5, respectively, adopted from De Leeuw et al. (2021)) were utilized for the Raikoke eruption. In EMAC, the StratProfile has been applied in the RD1SD-raik-02 (raik-02) sensitivity simulation, whereas the VolRes1.5 injection profile was used in the RD1SD-raik-03 (raik-03) simulation. In both simulations the SO$_2$ release is approximately 1.5 Tg (SO$_2$) (1.57 for raik-02 and 1.5 for raik-03) of SO$_2$ into the atmosphere, the only difference lies in the vertical distribution. In raik-02

(StratProfile profile from De Leeuw et al. (2021)), 69% of the volcanic SO$_2$ mass (1.09 Tg) is emitted into the stratosphere, with the primary peak occurring at 12-13 km altitude. In contrast, for raik-03 (equivalent to the VolRes1.5 profile in De Leeuw et al. (2021)), only 43% of the SO$_2$ mass (0.64 Tg) is emitted into the stratosphere, with the primary peak located around 10 km altitude in the upper troposphere. Additionally, raik-04 is based on the setup from raik-02 with additional emissions stemming from the Ulawun volcano. Table 5 lists the set-ups applied in all sensitivity simulations.

**Table 5.** Input parameters of the three sensitivity simulations used in this study. The table lists the prescribed volcano emissions for each sensitivity simulation, with their eruption time and the injected SO$_2$ mass into the stratosphere and into all model layers.

| Sensitivity simulation | Volcano | SO$_2$ emission date in 2019 | Stratospheric emission altitude [km] | Emitted SO$_2$ mass into the stratosphere [Tg] | Total emitted SO$_2$ mass into all layers [Tg] |
|---|---|---|---|---|---|
| raik-02 | Raikoke | 21-22 June | 12-13 | 1.09 | 1.57 |
| raik-03 | Raikoke | 21-22 June | 12-13 | 0.64 | 1.5 |
| raik-04 | Raikoke | 21-22 June | 12-13 | 1.09 | 1.57 |
| | Ulawun | 26 June | 16-19 | 0.14 | 0.14 |
| | Ulawun | 3-4 August | 11-15 | 0.2 | 0.2 |





For a detailed study focusing solely on the $SO_2$ mass burden originating from volcanic eruptions, the results of the RD1SD-base-01 reference simulation have been subtracted from the sensitivity simulations. This approach isolates the $SO_2$ mass specifically attributable to volcanic activity from other anthropogenic or outgassing volcanic emissions, thereby enabling a more precise analysis of its impact. Figure 4 illustrates the $SO_2$ mass emitted at different altitudes in both simulations. The altitude of the $SO_2$ emissions significantly influences their atmospheric distribution and dispersion patterns, impacting their climate effects, lifetime, and oxidation rates (Höpfner et al., 2015).

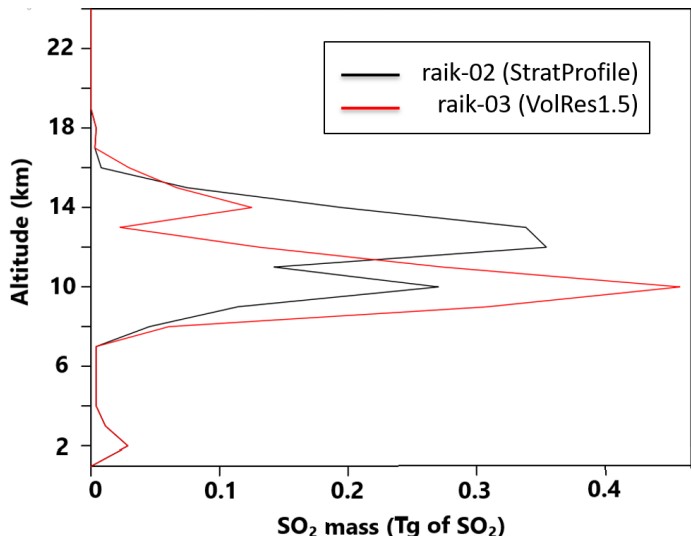

**Figure 4.** Shown is the estimated total emitted $SO_2$ mass for the Raikoke eruption in 21 and 22 June 2019 for two different EMAC set-ups. In the first one, represented by raik-02 (black line), most of the $SO_2$ mass (69%) is emitted into the stratosphere (De Leeuw et al., 2021). Conversely, in the second set-up (raik-03,red line), most of the $SO_2$ mass (57%) is emitted into the troposphere (De Leeuw et al., 2021).

The combined $SO_2$ mass in the troposphere and stratosphere represents the total $SO_2$ mass burden from the Raikoke eruption. This mass burden is then compared with the global $SO_2$ mass burden derived from the TROPOMI/Sentinel-5P satellite after the Raikoke eruption and until mid-July 2019 (De Leeuw et al., 2021) (see Figure 5). The raik-02 simulation (red line), which assumes a larger emission of $SO_2$ into the stratosphere, aligns more closely with TROPOMI's derived $SO_2$ data than the raik-03 simulation (orange line), in which a larger proportion of $SO_2$ is released into the troposphere.

Both simulations accurately capture the $SO_2$ mass burden peak at approximately 1.8 Tg. The peak values of the EMAC simulated Raikoke $SO_2$ mass are slightly larger than the total emissions presented in Table 5, due to the application of AK_15km to the EMAC results. Moreover, raik-02 (red curve) shows a better long-term agreement with TROPOMI estimates than raik-03 (orange curve), however both consistently remaining within the estimated uncertainty range (as drived by Theys et al. (2017)). For TROPOMI, uncertainties of $SO_2$ in the stratosphere are approximately ±30% of the retrieved VCDs (Theys et al., 2017). Conversely, the raik-03 simulation exhibits a more rapid decline of the $SO_2$ mass in the stratosphere, compared to the





TROPOMI data after the Raikoke eruption, suggesting a faster removal of $SO_2$ from the atmosphere, with an exception for the first two days after the $SO_2$ mass peak, where raik-03 agrees better with the measurement based estimates than raik-02.

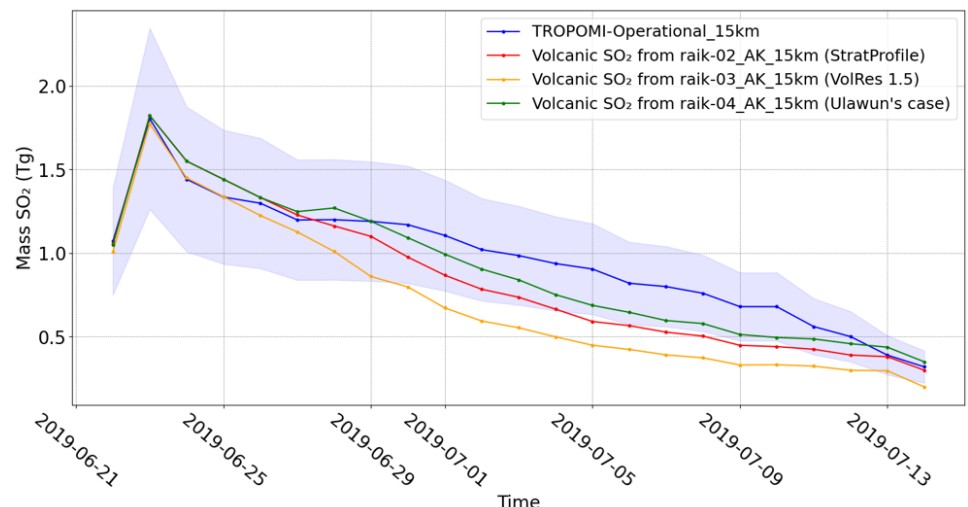

**Figure 5.** The daily evolution of the global $SO_2$ mass (Tg of $SO_2$) after the 2019 Raikoke and Ulawun volcanic eruptions retrieved from TROPOMI and different EMAC sensitivity simulations. raik-02 and raik-03 denote simulations with $SO_2$ mass from Raikoke emitted mostly in the stratosphere and troposphere, respectively. raik-04 is based on raik-02 with additionally emitted $SO_2$ mass into the stratosphere originating from the Ulawun volcano. The blue shading represents the uncertainty estimate for the TROPOMI product.

The discrepancies between TROPOMI $SO_2$ based estimates and EMAC simulation results can be attributed to the vertical
injection profiles (which might deviate from reality) or to the rate of sulfur removal from the atmosphere. Cai et al. (2022) suggest that additional injections are required after the initial Raikoke plume to accurately replicate the observed $SO_2$ mass, underscoring the complexity of modeling volcanic $SO_2$ emissions and their interactions in the atmosphere. To address these differences, a sensitivity simulation labeled raik-04 (green curve in Figure 5) was conducted. It is based on the setup of the raik-02 simulation, chosen because it best matches the temporal evolution of TROPOMI derived $SO_2$ mass, but additionally
includes emissions from the Ulawun volcano in the Southern Hemisphere. The Ulawun eruptions on 26 June 2019, at 12:00 UTC and 3 August 2019 at 12:00 UTC were taken into account, with each eruption lasting six hours. During the first eruption, 0.14 Tg of $SO_2$ was injected at altitudes between 16 and 19 km in the model. For the second eruption, 0.2 Tg of $SO_2$ was emitted at altitudes between 11 and 15 km. As shown in Figure 5, these adjustments improved the temporal evolution of $SO_2$ mass, slowing the decline in the raik-04 simulation following the Ulawun emission injections on June 26 due to the increased
$SO_2$ mass in the stratosphere.

During the Raikoke eruption and up to 29 June 2019, all simulations consistently show a continuous decrease of the $SO_2$ mass. TROPOMI data indicates a decline rate of approximately 0.08 Tg($SO_2$)/day, which is slower than the decline simulated by the EMAC model. The raik-03 simulation shows a decline rate of around 0.14 Tg($SO_2$)/day, indicating a lower $SO_2$ mass compared to TROPOMI. Conversely, the raik-02 simulation exhibits a slower decline rate of 0.1 Tg($SO_2$)/day. However, raik-





04 aligns most closely with TROPOMI, with a decline rate of 0.09 Tg(SO$_2$)/day. This closer match can be attributed to the additional Ulawun emissions injected into the stratosphere on 26 June 2019, which increases the mass of SO$_2$ in the atmosphere. Note that for the first two days following the SO$_2$ mass peak, TROPOMI observations indicate a rapid decline rate of approximately 0.15 Tg(SO$_2$)/day. This rate aligns most closely with the raik-03 simulation, where a significant amount of SO$_2$ is emitted at lower altitudes, resulting in a decline rate of 0.14 Tg(SO$_2$)/day. In contrast, the raik-02 and raik-04 simulations, which involve the majority of SO$_2$ being emitted into the stratosphere, exhibit a slower decrease of about 0.12 Tg(SO$_2$)/day. Between 29 June and 15 July 2019, all EMAC simulations show a decrease similar to TROPOMI, at a rate of approximately 0.05 Tg(SO$_2$)/day.

Over the period from 22 June to 15 July 2019, raik-04 aligns most closely with TROPOMI observations, by simulating about 3% lower SO$_2$ mass than TROPOMI over the entire period. raik-02 indicates a mean relative difference of 10%, while raik-03 simulates lower values than TROPOMI with a mean relative difference of 25% due to differences in the decline rate and SO$_2$ vertical injection profile. The fact that all EMAC simulations fall within the 30% uncertainty range of the SO$_2$ total column in TROPOMI provides confidence that EMAC correctly captures the main processes required to represent SO$_2$ oxidation and depostion.

To summarize, this analysis shows the capability of all sensitivity simulations, to reproduce the TROPOMI measured peak after the Raikoke and Ulawun eruptions. Furthermore, the consistent decay rates between TROPOMI data and sensitivity simulations, particularly raik-04 (which encompasses both, the Raikoke and Ulawun eruptions in the stratosphere), as well as accounting for the deposition of most of the initially emitted SO$_2$ mass within EMAC, further underline the model's realism in capturing the intricate processes of SO$_2$ emission, oxidation, and deposition associated with volcanic eruptions. Nevertheless, over extended durations, various factors such as simulated wind patterns, radiative heating effects, and mixing dynamics can introduce deviations between model results and real-world observations. These complexities highlight the ongoing challenges in achieving complete concordance between model simulations and empirical data over prolonged temporal scales.

### 5.3 Evaluation of SO$_2$ from anthropogenic and outgassing volcano emissions

EMAC SO$_2$ emissions from both, anthropogenic and outgassing (non-eruptive) volcanic sources, are derived from prescribed emission inventories, specifically CMIP6 and the AeroCom Project, respectively. Each of these emission inventories is based on distinct assumptions that may not accurately reflect the actual emitted SO$_2$ masses and injection heights. Consequently, only a relative comparison of SO$_2$ hotspots (i.e. with large SO$_2$ emissions) and background regions between the EMAC model results and TROPOMI observations is feasible. Therefore, for the comparison discussed in this section, only the ratios and the relationship between EMAC results and retrieved TROPOMI SO$_2$ VCDs are investigated.

Table 6 lists the ratios between SO$_2$ VCDs simulated with the EMAC model compared to those retrieved from TROPOMI, around specific outgassing volcanoes, presented as a yearly mean for 2019. The table indicates that EMAC SO$_2$ VCDs are generally larger than TROPOMI values over volcanic regions. Note that, when comparing the satellite data to model results folded with the AKs, COBRA data reveals an uncertainty of approximately 27% to 32% on the retrieved SO$_2$ column, mainly due to instrumental noise (Theys et al., 2022).





**Table 6.** Ratios of SO$_2$ VCDs between EMAC results and TROPOMI retrievals over different volcano types in 2019. The in EMAC emitted SO$_2$ in molec /m$^3$/s at different altitudes, between 989 hPa and 577 hPa, are listed.

| Volcanoes | SO$_2$ emissions at heights [hPa] in molec /m$^3$/s | | | | | | SO$_2$ VCD in EMAC divided by SO$_2$ VCD in TROPOMI |
|---|---|---|---|---|---|---|---|
| | 577hPa | 746hPa | 845hPa | 926hPa | 966hPa | 989hPa | |
| Etna (37.7°N, 14.9°E) | 1.46e14 | 9.62e14 | — | — | — | — | 7 |
| Trajumulco (15°N, 91.9°W) | 6.84e13 | 1.29e14 | — | — | — | — | 6 |
| Mt Fuji (37.3°N, 138.7°E) | 2.74e12 | 4.81e12 | 4.85e11 | 1.12e15 | 2.85e14 | — | 5 |
| Nevado Ojos del Salado (27.1°S, 68.5°W) | 2.44e14 | — | — | — | — | — | 2.5 |

The comparison reveals notable discrepancies between the SO$_2$ VCDs over several volcanic regions, with EMAC-to-TROPOMI ratios of approximately 7 for Etna, 6 for Tajumulco, and 5 for Mt. Fuji. In contrast, a lower ratio of about 2.5 is observed over Nevado Ojos del Salado. Notably, these differences are out of the error margin of the satellite measurements, which typically ranges between 27% and 32%.

These differences are attributed to the volcanic emission inventory used within the EMAC model. The larger ratios derived for the first three volcanoes could be attributed to both, the SO$_2$ emission masses and the emission heights. In the AeroCom inventory (Dentener et al., 2006), the Etna, Tajumulco, and Mt. Fuji volcanoes are considered not only as outgassing, but also explosive volcanoes. The top of Etna is approximately 3300 meters, Tajumulco about 4000 meters, and Mt. Fuji about 3700 meters, with emissions reaching up to around 4500 meters (577 hPa). This indicates explosive volcanic activity, with emissions ranging from the "top of the volcano + 500 meters" to "top of the volcano + 1500 meters", as detailed in Sect. 3.2. Conversely, the volcano in Chile is categorized solely as an outgassing volcano, thus showing the lower ratio between EMAC results and TROPOMI retrievals. Continuously outgassing volcanoes, within the used model setup, emit from "the height of the volcano * 0.67" up to the height of the volcano.

For anthropogenically influenced regions, the EMAC results also shows larger SO$_2$ VCDs compared to those from TROPOMI, but with smaller ratios in most regions compared to volcanic areas. Figure 6 shows a geographical map depicting the selected areas which heve been investigated quantitatively, and Table 7 lists the ratio of EMAC SO$_2$ VCDs to TROPOMI VCDs in these SO$_2$ background and hotspot regions. In background regions, a specific area in central Africa (coordinates: 12°N, 15°E to 2°N, 27°E, respectively) and the South Atlantic Ocean (coordinates: 20°S, 20°W to 30°S, 5°W) reveal small discrepancies between the SO$_2$ VCDs (10-20%). Larger differences are derived in anthropogenically influenced regions: Europe (1.6 factor, 60% difference), the USA (1.8 factor, 80% difference), and India (2.5 factor, 150% difference). In Northeastern China the factor is 3.2 (220% difference), while in Southeastern China it is 2 (100% difference).




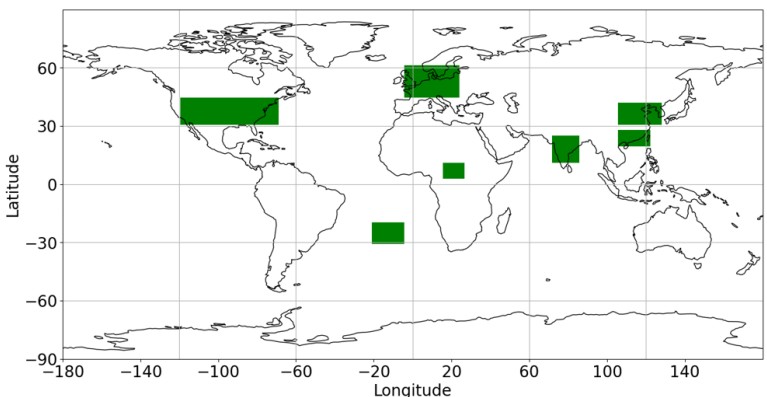

**Figure 6.** Geographical map showing the regions selected for this study.

**Table 7.** Ratios between $SO_2$ VCDs from EMAC results and TROPOMI retrievals in different background and $SO_2$ emission hotspot regions in 2019.

| Study regions | Ratios between $SO_2$ VCD in EMAC for standard case and $SO_2$ VCD in TROPOMI for standard case |
|---|---|
| South Atlantic Ocean | 1.1 |
| Africa | 1.2 |
| Europe | 1.6 |
| USA | 1.8 |
| Northeastern China | 3.2 |
| Southeastern China | 2.0 |
| India | 2.5 |

The regional discrepancies of $SO_2$ VCDs between EMAC and TROPOMI can be attributed to the large $SO_2$ emissions in these regions, originating from the CMIP6 emission inventory, as well as to their different original spatial resolutions. EMAC, with a coarse resolution of 300 km × 300 km, averages emissions over larger areas, potentially underestimating localized $SO_2$ peaks from sources like volcanoes or power plants. In contrast, TROPOMI/Sentinel-5P, with a much finer resolution of 3.5 km × 7 km, captures fine-scale variations. This difference might lead to discrepancies, especially in regions with strong

emissions, since the emissions in the model become instantaneously diluted by spreading the emitted mass over the model grid-boxes. For instance, in areas like India and China, where large $SO_2$ emissions take place, the ratios between EMAC and TROPOMI are larger, while background regions such as Africa show minimal differences in 2019, falling within TROPOMI's uncertainty range. In regions like Europe, the USA, and Southeastern China, which have lower $SO_2$ emissions than China and India for example, there is a better agreement between the two datasets.




Since TROPOMI only provides total VCD values, a detailed analysis of the vertical profile between both datasets is not possible. Therefore, it is difficult to ascertain whether the differences originate near the surface or higher up in the atmosphere. To address this, a comparison of the simulated $SO_2$ concentrations at the Earth's surface is conducted in the next section.

## 6    Evaluation of simulated $SO_2$ with ground-based measurements

To complement the inter-comparison of EMAC results with VCDs derived from TROPOMI data, we next compare the EMAC
results with ground-based measurements from observation networks in major $SO_2$-emitting regions worldwide. For this comparison, it is necessary that both datasets are aligned on the same latitude-longitude grid, as explained in Appendix A4. Specifically, the analysis centers on $SO_2$ concentrations and sulfur deposition fluxes over the USA, Europe, and at selected observational stations in China and Japan. These regions are chosen due to the availability of extensive and reliable datasets covering a two-decade period, from 2000 to 2019.

It is important to note that for the time series analysis in this section, mean/average values and standard deviation across stations are calculated for each year for both, the EMAC model results and the data from observational networks. The corresponding calculations and formulas can be found in the Appendix B.

     The results are presented in a aggregated form, i.e. we map the station data onto the model grid.

### 6.1    Sulfur concentration and deposition in the USA

For the United States, sulfur species simulated with EMAC near the Earth's surface (i.e. the lowermost grid box) are compared with observation data obtained from the CASTnet network. As detailed in Appendix A3.1, CASTnet provides surface-level observations, including monthly and yearly mean $SO_2$ concentrations in $\mu g/m^3$, and sulfur wet deposition fluxes in $kg(S)/ha$ per year. The $SO_2$ concentrations and sulfate amounts in precipitation samples are measured, whereas the dry deposition fluxes are simulated based on a multi-layer model.

Figure 7 shows the $SO_2$ concentration measured at the CASTnet sites (right panel, as mentioned above aggregated onto the model grid) and the EMAC simulated concentration (left panel). In both cases, 20 year averages are calculated. It is important to note that the EMAC results are only shown for grid boxes, where observational stations are located, which explains the presence of "empty boxes" in the EMAC model results. The figure indicates that Eastern USA sites exhibit larger $SO_2$ concentrations compared to the sites in the Western region in both datasets. This disparity is attributed to the higher density of $SO_2$ emission
sources in the Eastern USA compared to the Western part. This is also reported by Hardacre et al. (2021) and Qu et al. (2019).

     As shown in Figure 7, CASTnet (right panel) measures approximately 7 $\mu g/m^3$ at some individual sites in the Eastern USA, with other Eastern sites showing very low $SO_2$ concentrations of about 0.5 $\mu g/m^3$. On the left panel in Figure 7, the RD1SD-base-01 simulation driven by the CMIP6 inventory indicates that $SO_2$ concentrations at some individual Eastern sites are lower than those reported by CASTnet. However, on average, EMAC results show overall consistent $SO_2$ concentrations
between 1.5 $\mu g/m^3$ and a maxima of about 5.5 $\mu g/m^3$ at Eastern sites. In the Western region, both datasets show lower $SO_2$





concentrations, averaging around 1 $\mu g/m^3$ across all sites. However, the RD1SD-base-01 simulation results in larger SO$_2$ concentrations reaching up to 3 $\mu g/m^3$ at some Western sites.

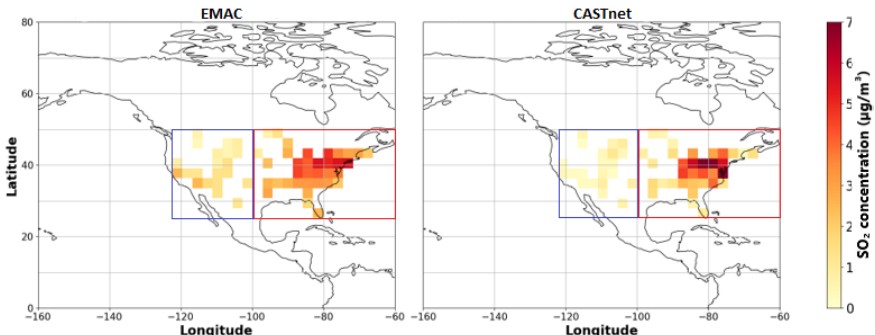

**Figure 7.** Geographical distribution of mean SO$_2$ concentrations for the years between 2000 and 2019 at the Earth's surface as simulated with EMAC (left) and observed at the CASTnet sites (aggregated onto the EMAC grid, right) in $\mu g/m^3$. The red and blue boxes indicate the regions, where the SO$_2$ emissions from the CMIP6 and EDGAR5 emission inventories are compared (details see text).

Figure 8 shows the comparison between both, the RD1SD-base-01 simulation results and CASTnet SO$_2$ concentrations as well as sulfur deposition fluxes across different regions in the USA. Specifically, SO$_2$ concentrations and sulfur deposition
fluxes from Western sites (panels (a) and (c), respectively), and from Eastern sites (panels (b) and (d), respectively), are shown. The comparison involves calculating the annual mean of SO$_2$ concentration in $\mu g/m^3$ and of sulfur deposition fluxes (wet and dry deposition) in $kg(S)/ha$ per year, averaged over Eastern, and Western USA sites, respectively. For both, surface SO$_2$ concentration and sulfur deposition flux, the RD1SD-base-01 simulation driven by the CMIP6 emission inventory effectively captures the decline in both regions of the USA for the period 2000-2019. As shown in Figure 8, the model tends to simulate
larger surface SO$_2$ concentrations than CASTnet by a factor of 2 in the Western region over the 20-year period (panel (a)), while showing approximately a factor of 1.2 larger SO$_2$ concentrations over the Eastern USA (panels (b)). For the western USA, EMAC shows decreasing surface SO$_2$ concentrations after 2000, which brings the simulated results into better agreement with the observations over time (see panel (a) in Figure 8). The large standard deviation derived from the datasets are attributed to the extensive dispersion of sulfur sources across a broad geographical area.
In the Eastern and Western USA, the largest part of sulfur removal occurs via wet deposition. This is effectively simulated by EMAC in agreement with CASTnet observations. For the calculation of deposition flux within the RD1SD-base-01 simulation, the deposited sulfate and SO$_2$ were converted to a sulfur equivalent. In panel (d) in Figure 8, EMAC shows a lower sulfur deposition flux over the Eastern USA, for wet deposition (10% lower EMAC values compared to CASTnet) and larger values for dry deposition (30% larger EMAC values compared to CASTnet) over the 20-year period. In the Western USA, EMAC also
simulates a 5% lower sulfur wet deposition flux compared to CASTnet over the entire 20-year period (panel (c) in Figure 8). Here, EMAC does not show lower values over all the time range, but indicates larger wet sulfur deposition between 2002 and





2008. For the sulfur dry deposition flux (orange lines), EMAC shows a factor of 2 larger values over the entire 20 year-period compared to CASTnet.

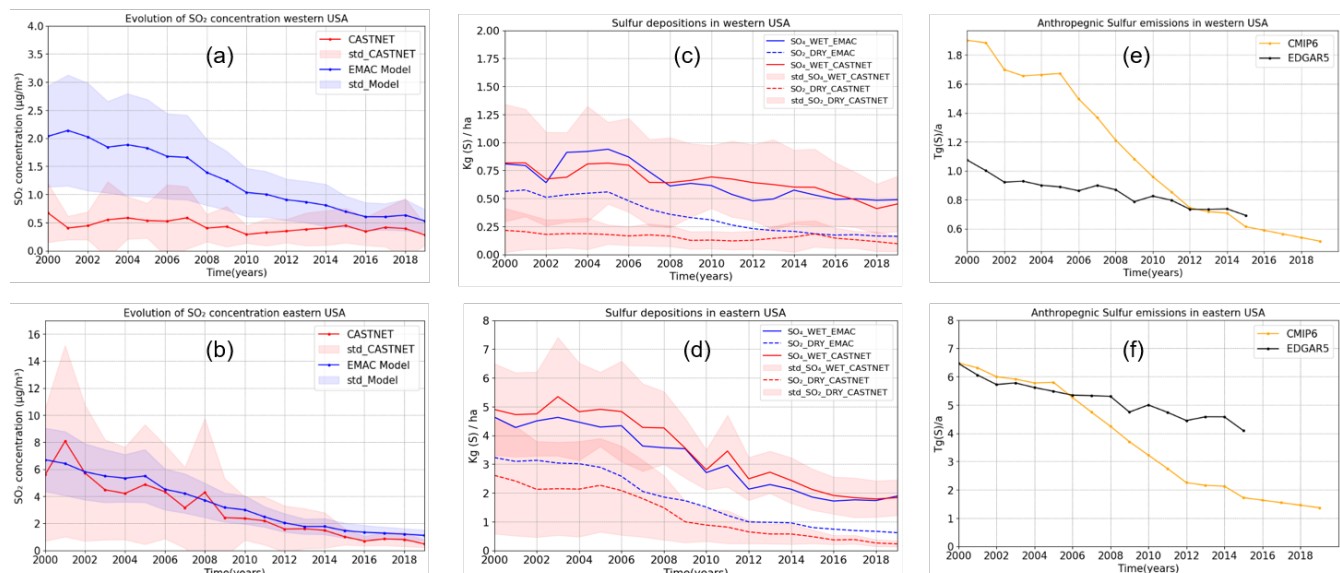

**Figure 8.** Time series of SO$_2$ concentrations from both, EMAC and CASTnet between 2000 and 2019 in the Western (panel (a)), and Eastern USA (panel (b)). The middle panels (c and d) show the evolution of wet and dry sulfur deposition fluxes between 2000 and 2019. For the calculation of deposition fluxes, the deposited sulfate and SO$_2$ were converted to sulfur equivalent. The right panels (e and f) show the comparison between the temporal evolution of CMIP6 and EDGAR5 anthropogenic sulfur emissions in Western and Eastern USA, as a yearly area integral between 2000 and 2019 (2015 for EDGAR5) of all the emission inventory grid boxes situated in the region marked by the blue and red boxes, respectively, in Figure 7.

Since the concentration simulated by the model is directly affected by the prescribed emissions, it is important to assess potential uncertainties of the applied emission inventory (CMIP6). For this, we present the comparison with the EDGAR5 emission inventory (see Sect. 3.2). The temporal evolution of sulfur emissions in Tg(S)/a from the CMIP6 emission inventory (orange lines in panels (e) and (f) in Figure 8) is compared with those from the EDGAR5 emission inventory (black lines in panels (e) and (f) in Figure 8). This comparison highlights the discrepancies and potential biases between the different emission inventories, which would be reflected in model results, if based on the alternative inventory. For the calculation of sulfur emissions, the emissions from SO$_2$ were also converted to sulfur equivalent. The anthropogenic emissions (particularly from fossil fuels, ship, road, and aircraft sectors) from both emission inventories are calculated as a yearly area integral over the Western USA region (see panel (e) in Figure 8), and over the Eastern USA region (see panel (f) in Figure 8). Both regions are shown in Figure 7, where the red and blue boxes represent the selected Eastern and Western USA regions, respectively.

For the western USA (panel(e) in Figure 8), the CMIP6 inventory has 50% larger sulfur emissions than the EDGAR5 inventory between 2000 and 2015. The picture is different for the Eastern USA (panel(f) in Figure 8), where the CMIP6





inventory indicates 10% less sulfur emissions than the EDGAR5 inventory between 2000 and 2015. The differences between the two emission inventories, particularly the larger anthropogenic sulfur emissions in CMIP6 compared to those in EDGAR5 over the Western USA, are a major factor contributing to the larger $SO_2$ concentrations simulated by EMAC compared to those observed at the CASTnet in Western USA (see panel (a) in Figure 8). Consequently, using the EDGAR5 emission inventory

over the Western USA would likely result in smaller $SO_2$ concentrations in the EMAC model, thereby reducing the bias between the CASTnet measurements and the EMAC results in that region.

For the final year of the study, 2019, the used CMIP6 emission inventory used for the RD1SD-base-01 simulation leads overall to a larger $SO_2$ concentration by a factor of approximately 1.6 compared to the CASTnet measurements.

## 6.2 Sulfur concentration and deposition in Europe

In Europe, 48 observational stations from the EMEP database are analyzed, as detailed in Appendix A3.2. First, the spatial distribution of $SO_2$ concentration over Europe from both datasets, is shown in Figure 9. EMAC results are only shown for grid boxes where observational stations are located, which explains the presence of "empty boxes" in the EMAC model results. Here, EMAC (left panel) shows the largest $SO_2$ concentrations (between 4 and 8 $\mu g/m^3$) over central East Europe, with lower $SO_2$ concentrations (between 0.3 and 3 $\mu g/m^3$) over sites in the United Kingdom and Western Europe. On the other side,

EMEP measures one very large $SO_2$ concentration (about 7.5 $\mu g/m^3$) over a grid box situated in Serbia, while showing lower $SO_2$ concentrations at the remaining sites (between 0.2 and 3.3 $\mu g/m^3$).

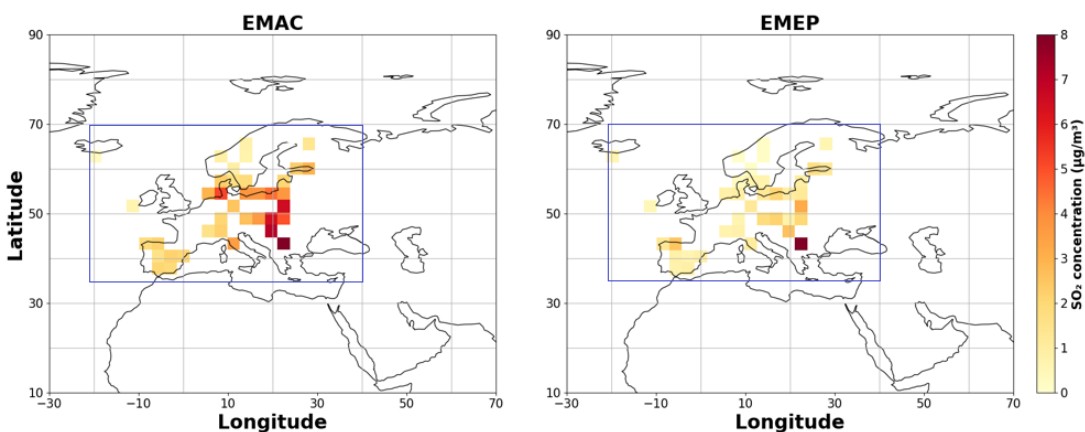

**Figure 9.** Geographical distribution of mean $SO_2$ concentrations for the years between 2000 and 2019 at Earth's surface as simulated with EMAC and observed at the EMEP sites in $\mu g/m^3$. The blue box indicates the region, where the $SO_2$ emissions from the CMIP6 and EDGAR5 emission inventories are compared (details see text).

Furthermore, Figure 10 illustrates the decline of $SO_2$ concentration across Europe, which is well captured by EMAC throughout the 20-year period (top left panel in Figure 10). Consequently, the temporal reduction in sulfur loss due to wet deposition is also accurately represented by EMAC (top right panel in Figure 10). However, the used CMIP6 emission inventory tends to



produce larger $SO_2$ concentrations within the RD1SD-base-01 simulation compared to observational data within the European domain (top left panel in Figure 10). Specifically, $SO_2$ surface concentrations from the model show a gradual decline between 2000 and 2012, with an annual decrease rate of 0.11 $\mu g/m^3$, whereas ground-based observational data indicate a slower reduction rate of 0.04 $\mu g/m^3$ per year during the same period. After 2012, both, model results and observational datasets, exhibit a more pronounced acceleration in the decline of $SO_2$ concentration, with rates of approximately 0.22 and 0.12 $\mu g/m^3$ per year,

respectively.

Over the entire 20-year period, EMAC driven by the CMIP6 emission inventory consistently shows larger annual mean surface $SO_2$ concentrations in Europe by a factor of approximately 1.8 compared to the EMEP dataset. Specifically, EMAC indicates larger $SO_2$ concentrations relative to observational data by a factor of 2 between 2000 and 2012, and with a lower factor of approximately 1.5 between 2012 and 2019. Regarding sulfur wet deposition flux, EMAC also simulates consistently

larger values than observed by EMEP by a factor of 1.3 over the entire 20-year period. Notably, a consistency is observed in the decline rates of both datasets, characterized by a yearly mean decrease of about 0.05 $\mu g/m^3$ throughout the duration from 2000 to 2019. Despite the differences in $SO_2$ concentration and sulfur deposition flux, it is noteworthy that the model exhibits a good alignment with observational data, as shown in the temporal progression of both, $SO_2$ concentration and sulfur deposition flux.

Similar to the study of sulfur emissions over the USA (see Sect. 6.1), the prescribed CMIP6 emission inventory in Europe used for the RD1SD-base-01 simulation, shows differences in the temporal evolution of emitted sulfur compared to the EDGAR5 emission inventory (see the low panel of Figure 10). Here, the anthropogenic sulfur emissions (particularly from fossil fuels, ship, road, and aircraft sectors) from both emission inventories are calculated as a yearly area integral between 2000 and 2019 (2015 for EDGAR5) over a region in Europe. The chosen region is marked by a blue box, as shown in Figure 9.

In the lower panel of Figure 10, an identifiable reduction of anthropogenic sulfur emissions across Europe is evident throughout the temporal evolution of both emission inventories. Specifically, the sulfur emissions in the CMIP6 inventory are 20% larger than those from the EDGAR5 inventory between 2000 and 2015 over Europe. This implies that using the EDGAR5 emission inventory would result in 20% less $SO_2$ being emitted in the model. Consequently, using the EDGAR5 emission inventory in Europe would likely result in lower $SO_2$ concentrations and therefore lower sulfur wet deposition fluxes in the EMAC model,

thereby reducing the bias between the EMEP measurements and the EMAC results.




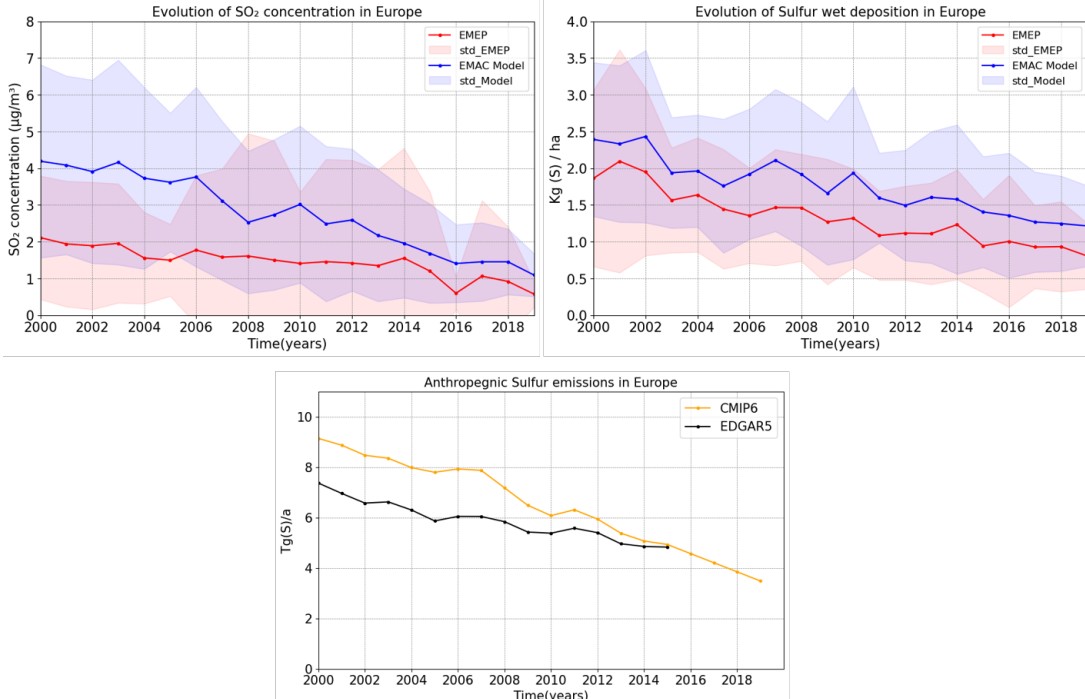

**Figure 10.** Temporal evolution of mean annual surface SO$_2$ concentration and sulfur wet deposition flux from EMAC (blue lines) and EMEP (red lines) between 2000 and 2019 at sites in Europe (top panels). The lower panel shows the comparison between the temporal evolution of CMIP6 and EDGAR5 anthropogenic sulfur emissions as a yearly area integral between 2000 and 2019 (2015 for EDGAR5) of all the inventory grid boxes situated in the region marked by the blue box in Figure 9.

For the final year, 2019, the used CMIP6 emission inventory used for the RD1SD-base-01 simulation leads to a larger EMAC SO$_2$ concentration by a factor of approximately 1.45 compared to EMEP measurements. Extrapolating the trend of the EDGAR inventory, we conclude that both inventories overestimate the European emission fluxes.

### 6.3 Sulfur concentration and deposition in China and Japan

For East Asia, a comprehensive investigation of SO$_2$ and its associated processes, particularly within China, is imperative due to the substantial contribution of Chinese SO$_2$ emissions, which account for 64–71% of the total emissions across Asia (Kuribayashi et al., 2012). However, as described in Appendix A3.3, it is complicated to find representative monitoring stations providing continuous, long-term datasets of measured SO$_2$ concentrations and sulfur deposition fluxes. Consequently, only 5 stations in Southeastern China and 9 over Japan from the EANET network have been selected for the present study. This

selection was based, as for the previous networks in the USA and Europe, on the availability of long-term measurements (2000 to 2019 for this study) providing both, measured SO$_2$ concentration in $\mu g/m^3$ and sulfate deposition fluxes in $mmol/m^2$



per year in EANET. The geographical distribution and SO$_2$ concentrations of selected stations in both, EMAC and EANET datasets at sites situated in China and Japan are shown Figure 11.

Here, both, EMAC and EANET datasets, indicate larger SO$_2$ concentrations at sites situated in China compared to those in
Japan. As shown on the right panel of Figure 11, EANET measures large SO$_2$ concentrations with a maximum of about 12 $\mu g/m^3$ at three sites in China, while showing lower SO$_2$ concentrations ranging between 1 and 3 $\mu g/m^3$ at the sites in Japan. On the left panel of Figure 11, the EMAC results show overall more consistent SO$_2$ concentrations with one site in China showing around 12 $\mu g/m^3$, while SO$_2$ concentrations at the other sites in China and Japan range between 2 and 8 $\mu g/m^3$.

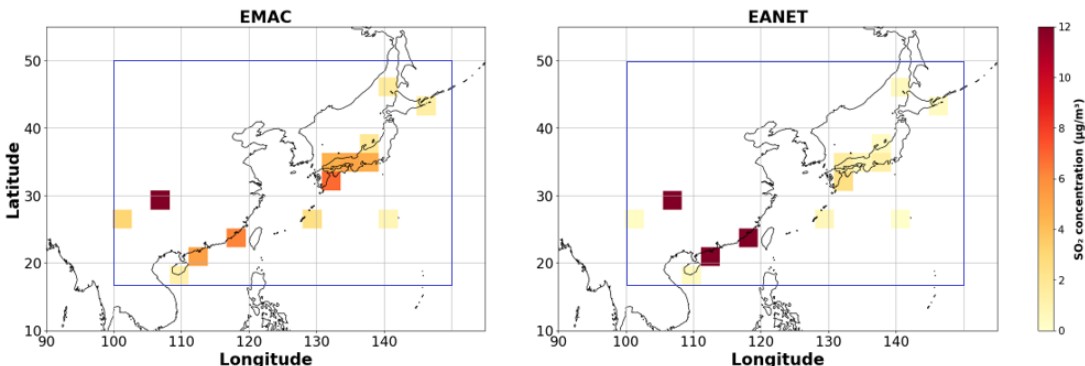

**Figure 11.** Geographical distribution of mean SO$_2$ concentrations for the years between 2000 and 2019 at Earth's surface as simulated with EMAC and observed at the EMEP sites in $\mu g/m^3$. The blue box indicates the region, where the SO$_2$ emissions from the CMIP6 and EDGAR5 emission inventories are compared (details see text).

Lu et al. (2010) and Ohara et al. (2007) reported a significant increase of SO$_2$ emissions in China during the early 2000s,
a trend confirmed by CMIP6 and EDGAR5 emission inventories, as depicted in the lower panel of Figure 12 (SO$_2$ emissions were converted to sulfur equivalent). This rise of sulfur emissions is also reflected in the SO$_2$ concentrations and the deposited mass flux of SO$_4^{2-}$, as illustrated in the same figure (top left and top right panels, respectively). EMAC indicates an overall increase of SO$_2$ concentration and the deposited mass flux of SO$_4^{2-}$ from 2000 till 2014. However, data from EANET reveals a different trend. According to the EANET network, SO$_2$ concentration increased until 2006 at a rate of 0.7 $\mu g/m^3$ per year.
On the other side, the deposited mass flux of SO$_4^{2-}$ also increased until 2006 at a rate of 0.11 $mmol/m^2$ per year. Afterwards, a decline of SO$_2$ concentration by 0.5 $\mu g/m^3$ per year and of the deposited SO$_4^{2-}$ mass flux by 1.3 $mmol/m^2$ per year until 2014, was measured.





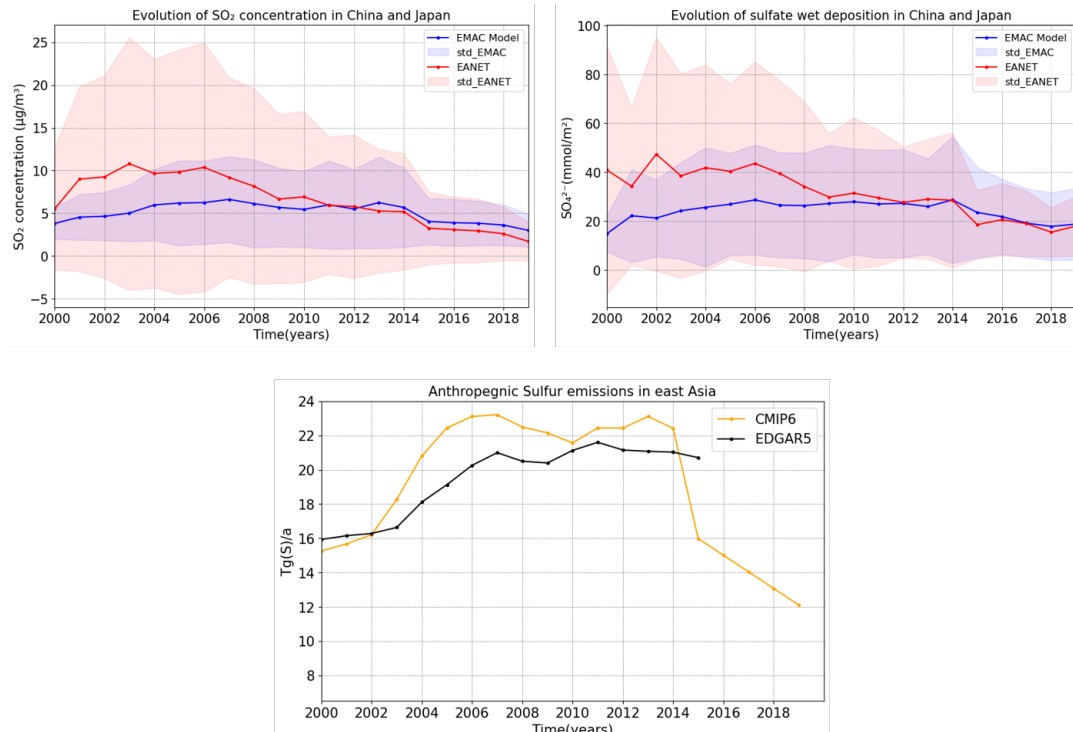

**Figure 12.** Temporal evolution of mean annual surface $SO_2$ concentration and sulfate wet deposition flux from EMAC (blue lines) and EMEP (red lines) between 2000 and 2019 at the sites in China and Japan (top panels). The lower panel shows the comparison between the temporal evolution of CMIP6 and EDGAR5 anthropogenic sulfur emissions as a yearly area integral between 2000 and 2019 (2015 for EDGAR5) of all inventory grid boxes situated in the region marked by the blue box in Figure 11.

Following 2014, numerous reports from environmental agencies and satellite observations have consistently indicated a significant decrease of China's $SO_2$ emissions. Studies by Wei et al. (2023) and Ronald et al. (2016) have highlighted that $SO_2$

emissions in China experienced a sharp decline post-2014, nearly vanishing by 2020. This reduction of $SO_2$ emissions after 2014 is also noticeable in the time evolution of sulfur emissions from the CMIP6 inventory utilized in this study (lower panel in Figure 12). From 2014 to 2019, a reduction exceeding 50% has been documented across the China-Japan region, with sulfur emissions decreasing from about 22 Tg(S)/a to 12 Tg(S)/a in the CMIP6 emission inventory. This decline of sulfur emissions notably influences the $SO_2$ concentration and the deposited $SO_4{}^{2-}$ mass flux over that region. Consequently, a remarkable

reduction, from about 5.5 $\mu g/m^3$ of $SO_2$ concentrations in 2014 to approximately 3 $\mu g/m^3$ by 2019, is detected (see the upper left panel in Figure 12). Additionally, a good agreement is also evident between EMAC results and EANET measurements regarding the evolution of deposited $SO_4{}^{2-}$ mass flux. Here, the deposited $SO_4{}^{2-}$ mass flux from both datasets indicate a decrease from 28 $mmol/m^2$ in 2014 to 18 $mmol/m^2$ in 2019 (see the upper right panel in Figure 12).

In contrast to Europe and the USA, a comparative analysis of $SO_2$ concentration and of the deposited mass flux of $SO_4{}^{2-}$

between the RD1SD-base-01- simulation results and measurements in China and Japan reveals noticeable differences in their





temporal evolution. The RD1SD-base-01 simulation results based on the CMIP6 inventory consistently exhibits lower $SO_2$ concentrations than EANET between 2000 and 2012, with a factor of approximately 1.5. Subsequently, larger $SO_2$ concentrations from EMAC of approximately 15% are detected in the years between 2012 and 2019. This pattern extends to sulfate deposition temporal evolution, where EMAC consistently simulates lower values than EANET from 2000 until 2014, with a

factor of approximately 1.6. However, from 2014 onwards, EMAC exhibits a reversal in trend, indicating larger values than EANET measurements by approximately 10% until 2019.

In the lower panel of Figure 12, the anthropogenic emissions from both, the used CMIP6 emission inventory and the EDGAR5 emission inventory are calculated as a yearly area integral over a region covering China and Japan. The chosen region is marked by the blue box shown in Figure 11. Again, the emissions from $SO_2$ were converted to sulfur equivalent.

In 2019, the used CMIP6 emission inventory used for the RD1SD-base-01 simulation leads to a $SO_2$ concentration of 3.6 $\mu g/m^3$ compared to 2 $\mu g/m^3$ measured by the EANET network, giving a bias of 1.6 $\mu g/m^3$ and a ratio of 1.8.

## 7   Conclusions and Outlook

The results of this study indicate that while the EMAC simulations demonstrate notable strengths in simulating the sulfur cylce, there are also areas that require improvement. Understanding the tropospheric sulfur budget forms the groundwork for the sub-

sequent analyses and examinations. This analysis is conducted using the EMAC RD1SD-base-01 model simulation, for which the CMIP6 $SO_2$ emission inventory was applied. The model demonstrates a closed sulfur budget, which has been compared with other results from literature. This assessment indicates a consistent representation of the model's sulfur chemistry, such as emissions, transport, chemical kinetics, and deposition. A closed sulfur budget allows for the evaluation of the EMAC model results against other observational data. By utilizing TROPOMI/Sentinel-5P measurements, the global distribution of natural

and anthropogenic $SO_2$ is identified in the EMAC model, showing different magnitudes compared to the $SO_2$ VCDs measured by the satellites instrument. Notably, the model indicates larger $SO_2$ VCDs, especially around regions with outgassing volcanic emissions. This discrepancy is attributed to the AeroCom emission inventory (Dentener et al., 2006) used within the EMAC model, which may not accurately reflect current outgassing volcanic activity. Given that the current emission inventory dates back to 2006, a new emission inventory for outgassing volcanic activities in the troposphere (Brühl et al., 2021) should be ap-

plied in future EMAC model setups. On the other hand, the enhanced $SO_2$ signals associated with the eruptions of the Raikoke and Ulawun volcanoes, as observed by the TROPOMI instrument, are successfully reproduced by the EMAC model when additional volcanic $SO_2$ emissions are explicitly included in the simulation setup. The temporal evolution of the additional global $SO_2$ mass is well reproduced by EMAC. Thus, future hindcast simulations with EMAC should also include a representation of eruptive volcanic emissions as proposed by Kohl et al. (2024).

Regarding the regions studied here, the biases calculated between the $SO_2$ VCDs from EMAC results and TROPOMI measurements and those between the EMAC simulated $SO_2$ concentrations and ground-based measurements over the USA, Europe and China and Japan in 2019, are consistent with each other. This strengthens the earlier hypothesis that the prescribed $SO_2$ emissions from the CMIP6 inventory used for the RD1SD-base-01 simulation might be overestimated over these regions, as





corroborated by a comparison with the EDGAR5 emission inventory. This underscores the importance of further investigating
the EMAC model results using various emission inventories to assess the range and sources of uncertainties, leading to a better
understanding of the behavior of $SO_2$ emissions across different regions.

Following our results, additional studies to further reduce the uncertainties of the knowledge about the atmospheric sulfur
budget are required. The selection of the emission inventory significantly influences the simulated $SO_2$ concentrations, which
consequently impacts the deposition processes. To enhance the understanding of these impacts, it is recommended to conduct
sensitivity simulations using various emission inventories and evaluate the model results against space-, air-, and ground-based
measurements. These simulations will help to quantify the uncertainties and variations associated with different inventories in
different regions, leading to more accurate simulated $SO_2$ concentrations and sulfur depositions.

Future research should focus on evaluating the simulated $SO_2$ VCDs using high-resolution satellite instruments. Unlike
the TROPOMI instrument, which provides a daily global coverage, the Geostationary Environment Monitoring Spectrometer
(GEMS) launched in 2020 (Kim et al., 2020), the Tropospheric Emissions: Monitoring of Pollution (TEMPO) instrument
launched in 2023 (Zoogman et al., 2017), and the Sentinel-4 instrument (Stark et al., 2013) are dedicated to measuring air
quality across Asia, North America, and Europe, respectively, every hour. The available high-frequency, near-real-time data
provide an excellent basis for validating the model's ability in simulating atmospheric $SO_2$ and capturing short-term variations
and transient events, such as pollution spikes and weather-related changes.





## Appendix A: Description of the used observational data

### A1 Satellite observations

In this study retrievals from the TROPOMI instrument on board the Sentinel-5P satellite are employed to investigate the $SO_2$ VCD. Sentinel-5P is the first Copernicus mission specifically designed for atmospheric monitoring, as mentioned by ESA (2017). Here, two dimensional level-2 products from the TROPOMI instrument are used. These products represent the original $SO_2$ data retrieved from the spectra observed by TROPOMI, including the geographical coordinates and resolution parameters such as scanline and ground pixel. These dimensions constitute the so-called "satellite orbit".

The retrieval data in TROPOMI is organized vertically into pressure layers from an a-priori profile of a CTM, namely the Tracer Model 5 (TM5) (Huijnen et al., 2010). In the case of $SO_2$, the data is divided into 34 distinct layers, varying approximately from the Earth's surface to 0.1 hPa (i.e. around 60 km).

In this study two distinct products are used:

1. The operational algorithm retrieves first the concentration of $SO_2$ integrated along the mean light, i.e., the so-called Slant Column Density (SCD).

   VCD cannot be directly measured from the satellite, thus, the conversion of the SCD into VCD becomes essential. This conversion process relies heavily on the air-mass factor "M":

$$VCD = \frac{SCD}{M},$$  (A1)

where the air-mass factor is calculated based on the formulation by Palmer et al. (2001), as follows:

$$M = \int m(p) \cdot s(p)\, dp.$$  (A2)

Here, m(p) is a weighting function reflecting the sensitivity of the satellite instrument to different altitudes. This function can be determined through pre-calculation or computational methods using a radiative transfer model. For the actual TROPOMI products, m(p) is given by the Linearized Discrete Ordinate Radiative Transfer (LIDORT) model, as introduced by Spurr et al. (2001). The term s(p) represents the vertical shape factor, which describes the normalized vertical profile of the $SO_2$ mixing ratio as a function of atmospheric pressure (Palmer et al., 2001). This profile can be obtained a-priori from any CCM or CTM. For instance, the CTM Tracer Model 5 (TM5) model is used as an a-priori profile for the TROPOMI/Sentinel-5P retrieveal. Therefore, Equation A2 could also be written as:

$$M_{\text{TM5}} = \int m(p) \cdot s_{\text{TM5}}(p)\, dp.$$  (A3)



2. The Covariance-Based Retrieval Algorithm (COBRA) product represents the latest advancement in $SO_2$ retrieval techniques from TROPOMI onboard the Sentinel-5 Precursor satellite (Theys et al., 2021).

Finally, it is important to note that the TROPOMI level-2 products are provided with the corresponding averaging kernels (AKs) for each case (Theys et al., 2017). These qualify the vertical sensitivity of satellite instruments and are important for ensuring a fair comparison with other types of data, especially atmospheric chemistry model simulation results (Veefkind et al., 2012). A detailed explanation of the AKs is provided in Appendix A2, together with an explanation of how the model data have been prepared for comparison with satellite measurements.

## A2 Post-processing of model data for comparison with satellite observations

For a global investigation of atmospheric sulfur chemistry within the EMAC model, a comparison of model results from the SORBIT submodel (Jöckel et al., 2010), with $SO_2$ products from satellite measurements (TROPOMI on board of Sentinel-5P in this study) is perfomed. TROPOMI $SO_2$ products are structured based on scanline and ground pixel, with the scanline representing the direction of the satellite's flight and the ground pixel indicating the resolution of the data. The EMAC model operates on a Gaussian lat-lon grid. To facilitate a meaningful comparison, the TROPOMI data must be regridded to match the grid of the EMAC model. This process involves reducing the fine resolution of TROPOMI to align with the coarser resolution of the model. However, before conducting the comparison, the model data needs to be folded with averaging kernels (AKs) from the retrievals to ensure its compatibility with TROPOMI data (Theys et al., 2022). The Averaging Kernel defines the sensitivity of the retrieved column, obtained from satellite-based measurements, to variations in the true profile of the measured trace gases based on a CCM or CTM (Rodgers, 2000). To properly weight the model data, it first needs to be brought onto the same horizontal and vertical grids as the AKs. This involves horizontally mapping the $SO_2$ mixing ratio from the SORBIT submodel onto the instruments grid resolution using the nearest neighbor method. Subsequently, a vertical linear interpolation is executed to align the 90 pressure levels of the simulated $SO_2$ mixing ratio with the 34 layers of the a-priori profiles used for the retrievals. Afterwards, the vertically interpolated $SO_2$ mixing ratio profiles are converted into a partial column for each of those grid-boxes (i.e. DU or molecules/cm$^2$). The vertically interpolated model data is then ready to be multiplied at each level with the corresponding averaging kernel and vertically integrated to yield the VCD. This step is important, as it translates the model $SO_2$ VCD into the signal that would be detected by the satellite. Finally, the VCDs of $SO_2$ from both, the model and the TROPOMI retrieval, are conservatively regridded from the instrument grid to the original EMAC latitude-longitude grid, and can be compared to each other.

Since the retrieved VCDs depend on simulated a-priori vertical profiles (represented as s(p) in Equation A2), which in turn depend on prescribed, mainly anthropogenic and volcanic $SO_2$ emissions, the COBRA dataset (see Sect. A1) provides four different VCDs for specific cases:

- The standard case (or "polluted case") is obtained using profiles of daily forecasts from the global CTM TM5 (Tracer Model 5, version TM5-chem-v3.0; Huijnen et al. (2010)).





- The 1 km case is obtained using 1 km thick box profile concentrating between the surface and 1 km (0 to 1 km), and representing a situation of passive degassing volcanoes or anthropogenic near-surface emissions.

- The 7 km case is obtained using 1 km thick box profile centered at 7 km (6.5 to 7.5 km), indicating a case of a moderate volcanic eruption.

- The 15 km case is obtained using 1 km thick box profile centered at 15 km (14.5 to 15.5 km), reflecting an explosive volcanic eruption case.

It is important to note that in order to compare the model VCDs with the four described VCDs cases from TROPOMI,
similar assumptions need to be adopted to ensure a valid comparison. By using the averaging kernel, one can ensure that the comparison between the satellite observations and the model output is meaningful, reflecting the same observational biases and sensitivities. In TROPOMI products, to conserve space, only the total column averaging kernel for the TM5 standard "pollution" case is provided as described by Theys et al. (2017):

$$AK(p) = \frac{m(p)}{M_{\text{TM5}}}, \tag{A4}$$

where $M_{TM5}$ represents the total air-mass factor of the vertical profile of the TM5 model and is calculated following Equation A3. Importantly, m(p) is consistent across all four cases, and the AK is calculated for the four distinct s(p) profiles. Consequently, we can easily recalculate the AK for each situation by scaling the polluted (or standard) averaging kernel by air-mass factor ratios $M_{\text{TM5}}/M_{\text{box}}$, as described by Eskes and Boersma (2003):

$$AK_{box}(p) = AK(p) \cdot \frac{M_{\text{TM5}}}{M_{\text{box}}}. \tag{A5}$$

Here, $p$ represents the pressure level at which the averaging kernel is stored for the TROPOMI product. $M_{\text{TM5}}/M_{\text{box}}$ serves as the scaling factor reported in TROPOMI products as "sulfurdioxide_averaging_kernel_scaling_box_{1,7,15}km".

In this study the averaging kernels are referred to as AK_polluted, AK_1km, AK_7km and AK_15km, and the resulting VCDs are resepctively expressed as, VCD_AK_polluted, VCD_AK_1km, VCD_AK_7km and VCD_AK_15km. Following, a detailed description of the different SO$_2$ VCD products used by TROPOMI, is given:

- VCD_AK_polluted is used for the standard "pollution" case. Here the COBRA product (see Sect. A1) is used due to its enhanced sensitivity to detect low SO$_2$ column densities. Additionally, a quality flag mechanism is employed to filter out potentially erroneous inputs such as cloudy pixels or missing values, which could deteriorate the results. Therefore, only data points with a quality assurance value above 0.5 (qa_value > 0.5) are considered reliable for analysis, as recommended by Theys (2023).



- VCD_AK_1km and VCD_AK_7km are also retrieved from the COBRA product. However, for volcanic activities, the prerequisite of the quality assurance value is no longer applicable. Instead, the only filtering criteria required is the SZA, where just data with SZA < 70° is considered (Theys, 2023). This can lead to higher signal-to-noise ratios in the satellite measurements, improving the quality of the data collected.

- The VCD_AK_15km product is applied specifically for eruptive volcano emissions. It is recommended to use the operational TROPOMI product (see Sect. A1) for such significant eruptions. Here, the filtering flag for volcanoes of SZA < 70° is similarly applied.

## A3  Ground-based measurements

For an inter-comparison with model results near the Earth's surface, ground-based measurements are used from three key sulfur-emitting regions: the United States of America (USA), Europe, and East Asia. These regions are selected because of their available and extensive datasets spanning a period of two decades, from 2000 to 2019. Next, a detailed explanation of the data is presented, with each region described separately.

### A3.1  USA

In the USA, data of various trace gases, including $SO_2$, are obtained from the CASTnet. This network (accessible at https://www.epa.gov/castnet, last accessed: 24 February 2024, Finkelstein et al. (2000)), provides surface-level observations including monthly and yearly mean $SO_2$ concentrations and sulfur deposition fluxes over the USA. In this study the data between 2000 and 2019 are used. Given the large size of the USA's land surface, a categorization of analyzed $SO_2$ has been undertaken, distinguishing between Eastern and Western sites. A total of 89 sites have been chosen for this study, as they represent data for both, $SO_2$ concentrations and sulfur deposition fluxes, over the two-decade period. Among these, 29 observation sites positioned West of 100°W longitude represent the Western USA, while the remaining 60 sites East of 100°W are representative for the Eastern regions. The spatial distribution of these site locations is presented in Figure A1.

In CASTnet, $SO_2$ and sulfate ($SO_4^{2-}$) concentrations are directly measured on a weekly basis at each of the stations. The concentration of sulfur compounds is multiplied by the volume of precipitation to calculate the deposition fluxes. This calculation provides the amount of sulfur deposited per unit area over a specific time period (in this study this is expressed in $kg(S)/hectares$ per year). However, measuring sulfur dry deposition fluxes faces some challenges, because it necessitates substantial instrumentation and technical resources (Hardacre et al., 2021). Therefore, deposition velocities are hourly estimated with the Multi-Layer Model (MLM, Meyers et al. (1998); Saylor et al. (2014)) and are integrated with measured $SO_2$ concentrations, land usage, and meteorological data to obtain the $SO_2$ dry deposition flux. The deposition velocity in the Multi-Layer Model (MLM) is based on the aerodynamic resistance, the quasi-laminar resistance to transport, and the surface uptake resistance (Baumgardner et al., 2002).



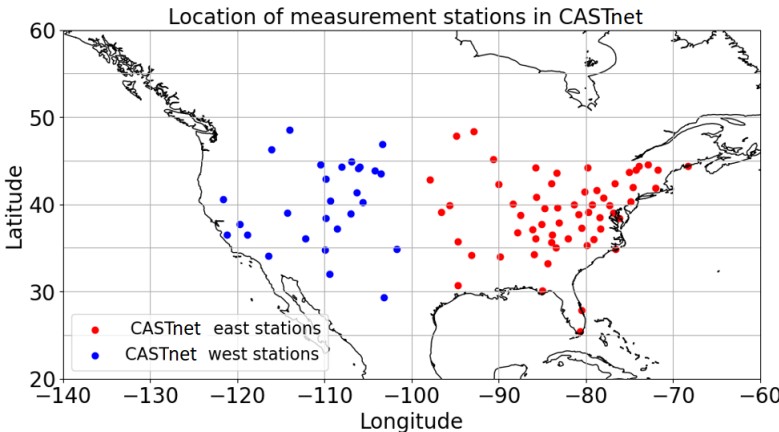

**Figure A1.** Map of the USA showing positions of Eastern (red points) and Western (blue points) CASTnet measurement sites used in this study.

### A3.2 Europe

The observational data from Europe offers extensive long-term atmospheric $SO_2$ measurements, obtained from the EMEP since 1972. This data repository, accessible via the EMEP database (http://ebas.nilu.no/, last accessed: 27 February 2024; Tørseth et al. (2012)), contains observations up to the present day. From EMEP, a total of 48 observational sites, distributed across Europe, are considered for our analyses. These sites not only monitor $SO_2$ concentrations, but also measure sulfate ($SO_4^{2-}$) amount in precipitation samples (Aas et al., 2019) ranging from 2000 to 2019. The concentration of sulfur compounds in the precipitation is multiplied by the volume of precipitation to calculate the deposition fluxes. This calculation provides the amount of sulfur deposited per unit area over a specific time period (in this study this is expressed in $kg(S)/hectares$ per year). Figure A2 visually illustrates the distribution of these observational sites across Europe.

$SO_2$ dry deposition data is, unlike to the CASTnet network, not available from the EMEP network. Consequently, the comparative analysis of sulfur deposition in Europe between observed data and the model results must rely solely on sulfate wet deposition from precipitation.

### A3.3 East Asia

In the East Asia region fewer observational stations were available compared to Europe and the USA. 14 stations were selected from the EANET, comprising 2 urban, 3 rural, and 9 remote locations in China (specifically Southeastern China) and Japan, as depicted in Figure A3. Notably, EANET stands out as the only network in East Asia equipped to monitor both, acid deposition and air pollution, with a particular emphasis on $SO_2$ (Ohizumi, 2023).

Same as for European measurement stations, $SO_2$ dry deposition fluxes are neither measured nor simulated from Asian networks. Fortunately, EANET provided access to both, yearly mean $SO_2$ near-surface concentrations and $SO_4^{2-}$ concentrations



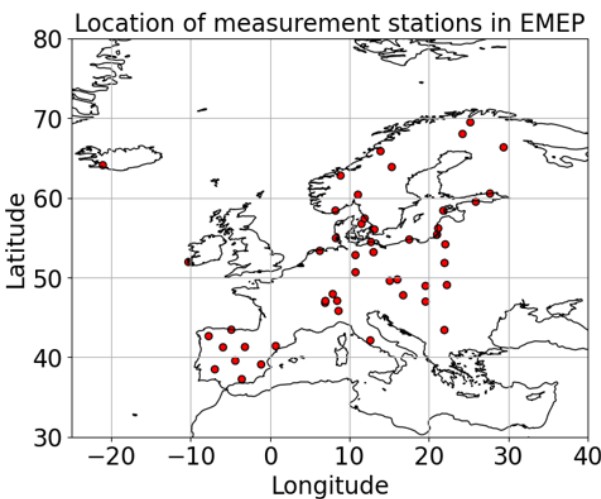

**Figure A2.** Map of SO$_2$ measuring sites from 2000 to 2019 in Europe from the EMEP network.

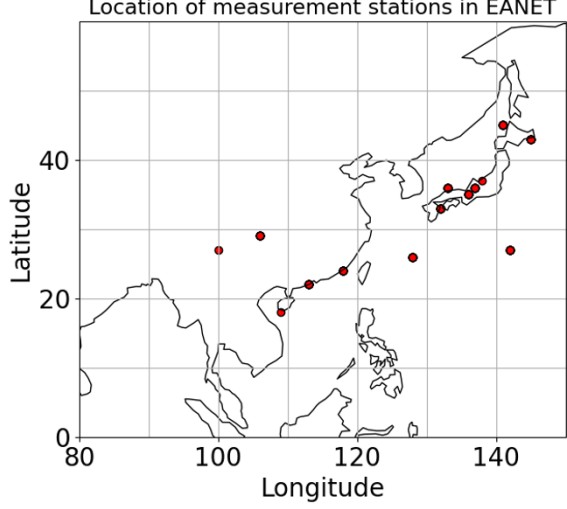

**Figure A3.** Map of SO$_2$ measuring sites situated in China and Japan from the EANET network.





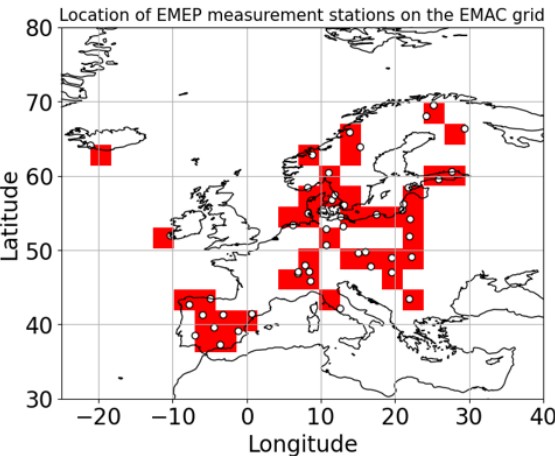

**Figure A4.** Visual representation of the original sulfur measurement sites in Europe (white circles) between 2000 and 2019 from EMEP network and after being mapped onto EMAC model's grid (red squares).

in precipitation. The concentration of $SO_4^{2-}$ in the precipitation is multiplied by the volume of precipitation to calculate the

deposition fluxes. This calculation provides the amount of $SO_4^{2-}$ deposited per unit area over a specific time period (in this study this is expressed in $mmol/m^2$ per year).

**A4   Post-processing of model and ground-based data for inter-comparison**

For this comparison, the measurement locations are aggregated to the $2.8° \times 2.8°$ horizontal grid of the RD1SD-base-01 EMAC simulation. Therefore, the nearest neighbor method is applied to assign the station measurement positions to the model grid.

Measurements from multiple stations within one grid-box are averaged. Figure A4 presents an example showing the original positions of the ground-based measurement stations in Europe and after being aggregated onto the model grid.

**Appendix B: Methodology for time series anlyses**

First, annual averages (2000-2019) for each measurement stations are caclualted from the monthly mean values. Next, these data are aggregated (with the nearest neighbor method) onto the model's grid for a direct comparison (see Sect. A4). For

EMAC grid boxes containing multiple observational stations, the mean value of all stations within that box is computed, in order to obtain a single representative value per grid box. Afterwards, the weighted mean over the grid boxes within a specific region is determined by summing all the grid box values weighted by the area of the grid boxes, as follows:

$$\mu_w = \frac{\sum_{i=1}^{N} w_i x_i}{\sum_{i=1}^{N} w_i}, \tag{B1}$$



where $\mu_w$ is the weighted mean, $x_i$ represents the value in grid box $i$, $w_i$ is the area (weight) of grid box $i$, and $N$ is the total number of the grid boxes. Consequently, the weighted standard deviation $\sigma_w$ is then expressed as:

$$\sigma_w = \sqrt{\frac{\sum_{i=1}^{N} w_i \left(x_i - \mu_w\right)^2}{\sum_{i=1}^{N} w_i}}. \tag{B2}$$

For the spatial analysis, the calculated mean values and the corresponding standard deviations over the entire 20 years, are calculated.

*Code availability.* The Modular Earth Submodel System (MESSy) is being continuously further developed and applied by a consortium of institutions. The usage of MESSy and access to the source code is licenced to all affiliates of institutions who are members of the MESSy Consortium. Institutions can become a member of the MESSy Consortium by signing the MESSy Memorandum of Understanding. More information can be found on the MESSy Consortium website (http://www.messy-interface.org, last access: 11 June 2025). The analysis presented here is based on model data published under DOI https://doi.org/10.26050/WDCC/ESCiMo2_RD1SD (Jöckel et al., 2024b) and at https://www.wdc-climate.de/ui/entry?acronym=DKRZ_LTA_853_dsg0002 (Jöckel et al., 2024a) and the sensitivity simulations have been performed with the code archived with DOI https://doi.org/10.5281/zenodo.15656328 (MESSy Consortium, 2025).

*Data availability.* The data of the RD1SD-base-01 simulation are available under the DOI https://doi.org/10.26050/WDCC/ESCiMo2_ RD1SD (Jöckel et al., 2024b) and at https://www.wdc-climate.de/ui/entry?acronym=DKRZ_LTA_853_dsg0002 (Jöckel et al., 2024a). The $SO_2$ data from the sensitivity simulations (RD1SD-raik-02, 03, 04) are accessible undr the DOI https://doi.org/10.5281/zenodo.15655676 (Jöckel, 2025). We acknowledge the use of the CASTNet database (https://www.epa.gov/castnet, last access: 12 June 2025, United States Environmental Protection Agency (2025)). The used EANET data could be found at https://monitoring.eanet.asia, last acces 12 June 2025. Information on the EMEP network can be found in Tørseth et al. (2012), and the data are available from http://ebas.nilu.no/, last access: 12 June 2025. The TROPOMI satellite data can be downloaded from the website (https://dataspace.copernicus.eu/explore-data/data-collections/ sentinel-data/sentinel-5p, last acces 12 June 2025).

*Author contributions.* IM, PJ and MD conceived the study, IM performed the data analysis, conducted the comparison between different datasets, and drafted the manuscript. PJ conducted the EMAC simulations. NT and JdL provided the satellite data and supported the application of the averaging kernels for comparison with the model reslts. JdL provided the estimated $SO_2$ emission fluxes of the Raikoke eruption. All authors contributed to the interpretation of the results and to the final version of the mansucript.



*Competing interests.* One of the co-authors is topical editor of the journal.

*Acknowledgements.* The EMAC simulations have been performed at the German Climate Computing Centre (DKRZ) through support from the Bundesministerium für Bildung und Forschung (BMBF). DKRZ and its scientific steering committee are gratefully acknowledged for providing the HPC and data archiving resources for this project ESCiMo (Earth System Chemistry integrated Modelling). We acknowledge G. Beachly (U.S. Environmental Protection Agency, EPA), C. Hardacre (Met Office, Exeter), H. Tost (Johannes Gutenberg University Mainz, Institute for Atmospheric Physics), and W. Aas (Norwegian Institute for Air Research, NILU) for providing the links to access the CASTNET, EMEP, and EANET ground-based measurement data. We are grateful to M. Kilian (LuftBlick OG) for the collaboration in developing a Python script used for comparing satellite observations with model simulation results. We also thank A. Schmidt (Deutsches Zentrum für Luft- und Raumfahrt, DLR) for providing the emission data of the Ulawun eruption and L. Bugliaro (DLR) for the internal review of our manuscript.

*Financial support.* This work is part of the PhD thesis by Makroum (2024), which was supported by the DLR MABAK (innovative methods for analyzing and assessing changes in the atmosphere and the climate system) project.



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
