# Peer review of "Evaluation of atmospheric sulfur dioxide simulated with the EMAC (version 2.55) Chemistry-Climate Model using satellite and ground-based observations"

_EGUsphere, 2025_

## Author Comment (AC1)

**Reply to reviewer #2**

5

15

20

Ismail Makroum 1, Patrick Jöckel 1, Martin Dameris 1, Nicolas Theys 2, and Johannes De Leeuw 3,4

Correspondence: Ismail Makroum (Ismail.makroum@dlr.de)

We thank reviewer #2 for her/his comments and the detailed evaluation of our paper. Below, we repeat each comment (in blue) and address it (in black). Changes of text in the manuscript are written in *italics*.

- This manuscript comprehensively evaluates sulfur dioxide in the EMAC chemistry-climate model. This is done by first verifying that the sulfur cycle is closed, meaning mass is neither lost nor gained. The SO2 depositions and vertical column densities are then evaluated with a set of ground-based and satellite observations. The structure of the manuscript is generally very well laid out and the methods are explained thoroughly and well executed. However, there are some areas where more concrete explanations are needed.

Thank you very much for this encouraging summary.

- P4, L90: The authors mention that the aerosol is not interactive in the simulations used for this study and refer to the aerosol radiative effects and heterogeneous chemistry. Is sulfate aerosol represented in some form in these simulations?
If so, please state for clarity. If not, what does this imply for the sulfur cycle?

This was indeed unlclear. In the revised manuscript we replaced the misleading sentcence by:

 $SO_2$  in the gas phase is oxidized by the hydroxyl-radical OH or directly photolysed. The major sink, however, is by transition into the aqueous phase, mainly cloud water, and further oxidation in the liquid phase. Since gaseous  $SO_2$  is not released on evaporation of cloud and rain droplets, the sulfur contents is in these cases transferred into a so-called residual (res) pseudo-aerosol tracer  $SO4_{\rm res,cs}$  with characteristics of a coarse mode soluble (cs) aerosol. This tracer is treated as aerosol tracer, e.g. by the sedimentaion submodel (SEDI), by the dry deposition submodel (DDEP), and by the wet scavenging submodel (SCAV). The details of the gas phase and liquid phase chemistry are documented as supplementary material.

- P16, Figure 5: Looking at this figure, the SO2 burden appears to decline exponentially in the TROPOMI retrieval, whereas all model simulations depict a more gradual, almost linear decay. The authors elaborate that the differences might stem from repetitive injections of SO2 into the atmosphere. However, the different changes in the decline rates suggest a difference in removal processes as well.

<sup>1Deutsches Zentrum für Luft- und Raumfahrt (DLR), Institut für Physik der Atmosphäre, Oberpfaffenhofen, Germany

<sup>2The Royal Belgian Institute for Space Aeronomy, Belgium

<sup>3The National Institute of Oceanography and Applied Geophysics, Italy

<sup>4The Abdus Salam International Centre for Theoretical Physics, Italy

Honestly speaking, we do not see this in Fig. 5. If there is a significant difference between the shape of declines at all, it is more the other way around: the model simulated declines are more exponentially shaped, whereas the TROPOMI retrieval based estimate is more linear in shape. Moreover, we do not have any indication that the sink processes represented in the model are not first order exponential, i.e. describable by an  $SO_2$  lifetime, which is determined mainly by washout, dry deposition, and oxidation with OH.

- Fig 7, 9, 11: The near surface SO2 concentrations in EMAC seem to be more spread out, is there an underestimation of initial removal close to the source or could it be related to how SO2 is emitted in the model or could the distribution be "flatter" due to e.g. numerical diffusion?

This is indeed a very good observation of our results. Indeed, presumably the rather coarse model resolution is the cause. Whereas  $SO_2$  in reality is in large parts emitted from point sources, in the model these emissions are instantaneously distributed over the entire grid-box. This is a well-known and common limitation of large-scale models for atmospheric chemistry.

**- Minor comments:**

- L6: close  $\longrightarrow$  closed
- 40 Done.

35

- L6-8: This sentence is a bit long and confusing

We split the sentence into two and reformulated: First, the tropospheric sulfur budget simulated by EMAC is verified to be closed. This closure means that all sulfur sources and sinks are balanced and no artificial gain or loss occurs over time due to numerical or conceptual errors.

- 45 L60: Jöckel et al. in parentheses
  - Done.
  - L89: isopren  $\longrightarrow$  isoprene
    - Done.
  - Table 4: Biomasse → Biomass
- 50 Done.
  - L204: citet  $\longrightarrow$  cited
    - Done.
  - L230: comparative → comparative
    - Done.
- 55 L365: heve  $\longrightarrow$  have
  - Done.

---

## Author Comment (AC2)

**Reply to reviewer #1**

Ismail Makroum1, Patrick Jöckel1, Martin Dameris1, Nicolas Theys2, and Johannes De Leeuw3,4

Correspondence: Ismail Makroum (Ismail.makroum@dlr.de)

We thank reviewer #1 for her/his comments and the evaluation of our paper. Below, we repeat each comment (in blue) and address it (in black). Changes of text in the manuscript are written in *italics*.

**1 General comments**

5 This manuscript is a thorough evaluation of the EMAC v2.55 sulfur simulations and makes a useful contribution by (i) closing a model-internal sulfur budget (ii) documenting how the model compares with satellite data in 2019 to evaluate how it responds to volcanic emissions and (iii) evaluating against long-term measurements, 2010–2019

The paper is well organized and generally clear. However, it is unnecessarily long, and the presentation of results is at times too detailed, making it difficult to extract the main messages and scientific significance. The description of the model setup partly repeats work published elsewhere, and it is not entirely clear what is new compared to earlier model versions (e.g., Jöckel et al., 2016).

Jöckel et al. (2016) describes our model setup and contribution to CCMI-1. Here, we analyse data from our contribution to CCMI-2022. This information is written in Section 3.1, lines 70-71 (original manuscript). However, since this is indeed an important information, we added to the abstract as well:

In this study, we present, for the first time, a comprehensive examination of atmospheric SO2 simulated by the EMAC model, here operated under the Chemistry-Climate Model Initiative (CCM-2022) protocol

For the interactive gas—particle chemistry, more detail would be beneficial, as the current description is incomplete for interpreting SO lifetime and deposition. I.e. the statement that "the simulation did not involve an interactive aerosol submodel" needs clarification. Does this mean that interactions with ammonia are excluded? If so, this should be explicitly stated, as ammonia strongly influences sulfur oxidation pathways, cloud pH, and the partitioning and deposition of sulfur.

This was indeed unclear. The interactions with ammonia are included in the liquid phase chemistry scheme. In the revised manuscript we replaced the misleading sentcence by:

<sup>1Deutsches Zentrum für Luft- und Raumfahrt (DLR), Institut für Physik der Atmosphäre, Oberpfaffenhofen, Germany

<sup>2The Royal Belgian Institute for Space Aeronomy, Belgium

<sup>3The National Institute of Oceanography and Applied Geophysics, Italy

<sup>4The Abdus Salam International Centre for Theoretical Physics, Italy

 $SO_2$  in the gas phase is oxidized by the hydroxyl-radical OH or directly photolysed. The major sink, however, is by transition into the aqueous phase, mainly cloud water, and further oxidation in the liquid phase. Since gaseous  $SO_2$  is not released on evaporation of cloud and rain droplets, the sulfur contents is in these cases transferred into a so-called residual (res) pseudo-aerosol tracer  $SO_{Tes,cs}$  with characteristics of a coarse mode soluble (cs) aerosol. This tracer is treated as aerosol tracer, e.g. by the sedimentaion submodel (SEDI), by the dry deposition submodel (DDEP), and by the wet scavenging submodel (SCAV). The details of the gas phase and liquid phase chemistry are documented as supplementary material.

**30 2 Specific comments**

40

45

50

- For the emissions it is important to notice that earlier global inventory for China have underestimated the reductions after around 2010. I am not sure if CMIP6 emissions have taken this into account? In Line 35 it is written that the emissions in China remain high, which is true, but the authors should also mention the substantial reductions in recent years.
   We added a reference analysing the recent reduction of SO2 emissions over China.
- Why is the comparison made with EDGAR 5 instead of the more recent EDGAR 6.1 (2024)? Using EDGAR 6.1 would enable comparison over the full 2000–2019 period. It would strengthen the analysis to run short sensitivity tests (1–2 simulations) using alternative inventories (e.g., EDGAR 6.1) to quantify whether biases in CMIP6 emissions explain the overestimation. If such tests are beyond the study's scope, this limitation should at least be discussed.
  - In our revised manuscript we added the more recent EDGAR 8.0 inventory to cover the full period. However, the interpretation of our model results and our conclusions do not change. Additional simulations with alternative emission inventories are indeed beyond scope, because the focus here is the evaluation of the CCMI-2022 model results (see also reply to General comments).
  - Why are you not comparing to the more recent EDGAR6.1 (from 2024) instead of EDGAR5? Then you would be able to compare it with the whole 2000-2019 period. It would have been useful to run the model with EDGAR and not only compare the emissions inventory to evaluate more directly if it biases in the CMIP6 emissions that cause the overestimation. Not necessarily a full rerun of the model but quantify the bias by re-running 1–2 short sensitivity test with different emissions. Though I understand if this is beyond the capacity of this work.
    - In our revised manuscript we added the more recent EDGAR 8.0 inventory to cover the full period. However, the interpretation of our model results and our conclusions do not change. Additional simulations with alternative emission inventories are indeed beyond scope, because the focus here is the evaluation of the CCMI-2022 model results (see also reply to General comments).
  - Sulfur data from Africa (INDAAF; https://indaaf.obs-mip.fr) and from Canada (CAPMoN) could have been included to provide a more complete global picture. In the U.S., CASTNET is responsible for air and aerosol data, whereas wet deposition data originate from the National Atmospheric Deposition Program (NADP). It appears NADP data were used,

60

65

70

75

80

85

In principle it would have been possible to include other regions in our analysis as well. However, in other regions, compared to the three regions USA, Europe, and East Asia/China, the network coverage is usually lower and time series of observations have gaps in the considered time range. For instance, according to our inspection, the INDAAF data contain only 4 stations with  $SO_2$  measurements, but none covering the persond 2000 - 2019, which was the criterion for our selection (see our Appendix A3 Ground-based measurements).

Moreover, we intentionally focused on the three regions since they are, on the one hand, the largest emitters in the industrial and energy sectors, and on the other hand, are observed by rather dense observational networks providing sufficiently long time series for evaluation.

For the USA we used the CASTnet data from the URL as indicated in our Appendix A3.1 USA. This data indeed comprises the wet deposition data from NADP/NTN. We added this information to the revised manuscript:

Dry deposition fluxes in this data set are derived with the Total Deposition Science Committee (TDep) Measurement Model Fusion method using the measured air concentrations from CASTnet and simualted CMAQ deposition velocities and fluxes. The wet deposition fluxes in this data set are estimated with the TDep Measurement Model Fusion method using the concentrations in precipitation and precipitation amounts measured by the National Atmospheric Deposition Program / National Trends Network (NADP/NTN).

Again, we selected only stations at which the measurements cover the two decades 2000 - 2019.

- The model response to volcanic emissions is evaluated only against TROPOMI satellite data. You write that "it is difficult to ascertain whether the differences originate near the surface or higher up in the atmosphere." It would strengthen the analysis to use in-situ data from 2019, which are available for that year and would allow a more direct comparison.
  Indeed, we tried this direct comparion, however the results were not conclusive. In-situ data are only aivailable near-surface, where the volcanic SO2 signal is weak or entirely absent, because the volcanic plume is transported and dilutet at higher altitudes.
- It is not written how trends were calculated, i.e. linear regression, Sen's Slope. Mann Kendall. This should be added to the methods.

Trends have been calculated with linear regression. This information has been added to the revised manuscript, Appendix B, and in the text.

For trend comparisons, it would be appropriate to refer to regional studies covering similar periods. Some suggestions: We thank the referee for pointing out the additional literature.

In Europe recent work done by EMEP: https://doi.org/10.4209/aaqr.230237. Seems like you have somewhat larger trend for SO2 (0.05 ug/m3/y) compared to EMEP (0.034 ug/m3/y). The difference may stem from site selection, trend method, or emissions used.

Unfortunately we cannot retrace the  $0.034 \,\mu \mathrm{gm^{-3}yr^{-1}}$ . Aas et al. (2024) report a  $\mathrm{SO}_2$  decline of  $-0.067 \,\mu \mathrm{gm^{-3}yr^{-1}}$  (see their supplementary Table S3) for the years 2000-2019, which is slightly larger than our derived trend. We added this information to the revised manuscript:

This trend, calculated by linear regression, is comparable to the trend of -0.067  $\mu gm^{-3}$  derived by Aas et al. (2024, see thier Table S3 for the perios 2000 – 2019), which has been derived with a different method and based on a slightly different number of stations.

Several studies in North America. Eg.: NADP data: https://doi.org/10.1016/j.atmosenv.2023.119783, CASTNet: https://doi.org/10.5194/acp-22-12749-2022 and from Canada by CAPMoN: https://doi.org/10.5194/acp-22-14631-2022

The comparison of our results with those of Conrad-Rooney et al. (2023) is limited, nevertheless we added to the revised text:

Conrad-Rooney et al. (2023) analysed the wet-deposition of  $SO_4^{-2}$  for urban and suburban sites separately, whereas we did not distinguish different site classes and analysed the total sulfur deposition. Thus, the results are not directly comparable. Nevertheless, Conrad-Rooney et al. (2023, see their Figures 2D and 5D) estimated for the year 2000 a wet deposition rate (urban stations) of approx.  $20 \text{ kg}(SO_4^{-2})\text{ha}^{-1}$ , equivalent to  $6.7 \text{ kg}(S)\text{ha}^{-1}\text{yr}^{-1}$ . For 2018, they derived a wet deposition rate of 5 (rural) to 7.5 (urban)  $\text{kg}(SO_4^{-2})\text{ha}^{-1}\text{yr}^{-1}$ , equivalent to 1.7 to 2.5  $\text{kg}(S)\text{ha}^{-1}\text{yr}^{-1}$ , respectively. These numbers agree, given the limitations of this comparison, with our results for the eastern USA (see Figure 8d, blue area).

W.r.t. to Benish et al. (2022) we added:

Benish et al. (2022) derived for the USA a decrease of the total sulfur deposition from  $5.3~{\rm kg(S)ha^{-1}}$  in 2002 to  $1.8~{\rm kg(S)ha^{-1}}$  in 2017 (see their Figure S14 (b)). These total fluxes are lower than the wet deposition fluxes derived by Conrad-Rooney et al. (2023) and than our results, but the derived trends are consistent.

Japan: https://doi.org/10.1016/j.envpol.2021.117842

We added:

90

95

100

105

110

115

120

In addition, Yamaga et al. (2021) found "no clear increase or decrease trends in the S deposition amounts throughout the 15-year period" (2003 - 2017) at 8 stations in Japan. This is also in line with our results (Figure 12, top right panel) due to the large standard deviations and because the overal decline is largely driven by emission reductions over China.

- China: https://doi.org/10.3390/su17198815

We added:

... This is an average decline of  $0.53~{\rm kg(S)\,ha^{-1}yr^{-1}}$  between 2014 and 2019 based on station data in South China and Japan. Xi et al. (2025) reported a China nationwide average decline rate (2013 – 2023) of -0.244

 $kg(S) ha^{-1}yr^{-1}$  and further showed that the trends in South and Central China are larger (negative) than this average (see their Figure 2). Thus, our results can be considered to be consistent.

**3** Technical corrections/spelling errors**

```
- Line 70: "caclulated" to calculated
            Done.
125
         - Line 80-81: "sun-synchrinously" to sun-
            Typo is corrected.
         - Line 89: "isopren to isoprene.
            Done.
130
         - Line 146: "histrotical" to historical.
            Done.
         - Line 149: "soley" to solely.
           Done.
         - Line 230: "comparatative" to comparative.
135
            Done.
         - Line 260: "correpsonding" to corresponding.
            Done.
         - Line 294: "drived" to derived
           Done.
         - Line 328: "deposition" to deposition
140
            Done.
         - Line 364: "heve" to have.
            Done.
         - Line 543; "cylce" to cycle
145
            Done.
         - Line 604: "retrieveal" to retrieval.
```

Done.

- Line 658: "resepctively" to respectively.
   Done.
- 150 Line 723: "anlyses" to analyses
  Done.
  - Line 724: "caclualted" to calculated
     Done.
  - Line 746 "undr" to under. Multiple instances of "acces" and "reslts" to be replaced with access and results.
  - Table 2 caption: "aersol" to aerosol.
     Done.

Done.

155

- Figure caption 12. EMEP should be replaced with EANET
   Corrected. Thanks for spotting this error.
- Standardize Tg(S)/yr vs Tg(S)/a
   We adapted all units to the exponential form as requested by the journal guidelines, i.e. Tg(S) yr-1 and similar.
  - There is a mix of units used for air concentration and fluxes (e.g., μg m³ vs mmol m² and kgS/ha). These should be standardized throughout the paper for clarity and comparability.
     Our intention was to use the units as reported with the corresponding measurements. Yet, we see the issue. Thus, in the
- revised manuscript we converted the deposition flux units mmol/m2 into kg(S) ha-1 in Section 6.3 to be consistent with Sections 6.1 and 6.2.

**References**

- Aas, W., Fagerli, H., Alastuey, A., Cavalli, F., Degorska, A., Feigenspan, S., Brenna, H., Gliß, J., Heinesen, D., Hueglin, C., Holubová, A., Jaffrezo, J.-L., Mortier, A., Murovec, M., Putaud, J.-P., Rüdiger, J., Simpson, D., Solberg, S., Tsyro, S., Tørseth, K., and Yttri, K. E.:

  Trends in Air Pollution in Europe, 2000–2019, Aerosol and Air Quality Research, 24, 230 237, https://doi.org/10.4209/aagr.230237, 2024.
  - Benish, S. E., Bash, J. O., Foley, K. M., Appel, K. W., Hogrefe, C., Gilliam, R., and Pouliot, G.: Long-term regional trends of nitrogen and sulfur deposition in the United States from 2002 to 2017, Atmospheric Chemistry and Physics, 22, 12749–12767, https://doi.org/10.5194/acp-22-12749-2022, 2022.
- Conrad-Rooney, E., Gewirtzman, J., Pappas, Y., Pasquarella, V. J., Hutyra, L. R., and Templer, P. H.: Atmospheric wet deposition in urban and suburban sites across the United States, Atmospheric Environment, 305, 119783, https://doi.org/https://doi.org/10.1016/j.atmosenv.2023.119783, 2023.
  - Xi, Y., Wang, Q., Zhu, J., Hao, T., Zhang, Q., Chen, Y., Tai, Z., Lin, Q., and Wang, H.: Nationwide Decline of Wet Sulfur Deposition in China from 2013 to 2023, Sustainability, 17, https://doi.org/10.3390/su17198815, 2025.
- Yamaga, S., Ban, S., Xu, M., Sakurai, T., Itahashi, S., and Matsuda, K.: Trends of sulfur and nitrogen deposition from 2003 to 2017 in Japanese remote areas, Environmental Pollution, 289, 117 842, https://doi.org/https://doi.org/10.1016/j.envpol.2021.117842, 2021.